# SMOC can act as both an antagonist and an expander of BMP signaling

J Terrig Thomas*, D Eric Dollins, Kristin R Andrykovich, Tehyen Chu, Brian G Stultz, Deborah A Hursh, Malcolm Moos

Division of Cellular and Gene Therapies, Office of Tissues and Advanced Therapies, U.S. Food and Drug Administration, Silver Spring, United States

**Abstract** The matricellular protein SMOC (Secreted Modular Calcium binding protein) is conserved phylogenetically from vertebrates to arthropods. We showed previously that SMOC inhibits bone morphogenetic protein (BMP) signaling downstream of its receptor via activation of mitogen-activated protein kinase (MAPK) signaling. In contrast, the most prominent effect of the *Drosophila* orthologue, *pentagone* (*pent*), is expanding the range of BMP signaling during wing patterning. Using SMOC deletion constructs we found that SMOC-ΔEC, lacking the extracellular calcium binding (EC) domain, inhibited BMP2 signaling, whereas SMOC-EC (EC domain only) enhanced BMP2 signaling. The SMOC-EC domain bound HSPGs with a similar affinity to BMP2 and could expand the range of BMP signaling in an in vitro assay by competition for HSPG-binding. Together with data from studies in vivo we propose a model to explain how these two activities contribute to the function of Pent in *Drosophila* wing development and SMOC in mammalian joint formation.

## Introduction

During development bone morphogenetic proteins (BMPs), comprising at least twenty structurally-related members of the transforming growth factor β (TGF-β) superfamily, are involved in many growth and differentiation events essential for determining body structure (*Wu and Hill, 2009*). Establishing temporospatial gradients or restricted distributions of BMP signaling is important for many of these processes, which are regulated by a number of mechanisms: ligand binding by extracellular BMP antagonists (*Brazil et al., 2015*), intracellular feedback inhibition downstream of the BMP receptor (*Massagué et al., 2005*), spatially restricted proteolytic processing (*Thomas et al., 2006*), and promotion or restriction of diffusion by interactions with extracellular matrix proteins such as collagen type IV (*Umulis et al., 2009*) and heparan sulfate proteoglycans (HSPGs) (*Matsumoto et al., 2010*; *Belenkaya et al., 2004*). In addition, BMP signaling is also influenced via communication with other signaling pathways, particularly those that act through mitogen-activated protein kinases (MAPKs). MAPK-directed phosphorylation of BMP receptor-regulated Smad 1/5/8 in the linker region inhibits BMP signaling by blocking Smad translocation to the nucleus (*Alarcón et al., 2009*; *Sapkota et al., 2007*).

We showed previously that SMOC, a matricellular protein associated with basement membranes (*Vannahme et al., 2002*) and expressed in developing brain, branchial arches, eye, pronephros, limb bud cartilage condensations, and joint interzones (*Okada et al., 2011*; *Rainger et al., 2011*; *Thomas et al., 2009*), inhibits BMP signaling (*Thomas et al., 2009*); following the addition of BMP2 to NIH3T3 fibroblasts transfected with SMOC, downstream phosphorylation of Smad 1/5/8 is blocked (*Thomas et al., 2009*). In *Xenopus* ectodermal explants (animal caps), SMOC was shown to activate MAPK signaling and inhibit Smad 1/5/8-mediated BMP signaling downstream of the constitutively active BMP receptor, BMPR1B (*Thomas et al., 2009*). Although the exact mechanism is not

*For correspondence: john.
thomas@fda.hhs.gov

Competing interests: The
authors declare that no
competing interests exist.

Reviewing editor: Hugo J
Bellen, Baylor College of
Medicine, United States

**eLife digest** During the development of an embryo, a group of proteins known as growth factors stimulate cells to divide and direct how organs and limbs form. One family of growth factors called bone morphogenetic proteins (BMPs) regulate the formation of bone and many other tissues in the embryo. BMPs are released from cells, diffuse away and are then detected by other cells. When BMPs attach to docking station-like structures on the cell surface, called receptors, they stimulate a signaling process inside the cell. In 2009, researchers found that a protein called SMOC blocks BMP activity in animals with backbones by triggering an interfering signal inside the cell. In flies, however, the equivalent protein can make BMP diffuse further from the cell that releases it.

To find out how SMOC can do both of these things, Thomas et al. – including some of the researchers involved in the 2009 study – conducted experiments to see which parts of SMOC are required to either block BMP signaling or encourage the diffusion of BMP. These experiments revealed that one end of SMOC can stick to molecules on the cell surface that are not receptors but are molecules where BMP can also bind. When this end of SMOC attaches to these sites, BMPs cannot bind and so diffuse further away.

Thomas et al. then produced complete or shortened versions of SMOC proteins to see how this affected BMP activity in frogs. These experiments indicated that the opposite end of SMOC is required for short-circuiting the BMP signal. The results also showed that, at lower concentrations, SMOC stimulates BMPs to diffuse, and that higher concentrations are required to block BMP signaling.

These findings suggest that similar to flies, SMOC can also stimulate BMPs to diffuse from the cell in animals with backbones. The next step will be to identify the cell surface receptor for SMOC to better understand the molecular mechanisms that inhibit BMP. The SMOC pathway could be targeted for therapeutic strategies to combat diseases associated with errors in BMP signaling like osteoarthritis, or in cell-based therapies where BMP signaling must be inhibited to produce cells needed to repair damaged tissues.

known, the activity was lost in the presence of non-phosphorylatable linker-mutant Smad, suggesting that BMP inhibition results from activation of MAPK signaling and subsequent ubiquitination and degradation of Smad following linker Smad phosphorylation (*Sapkota et al., 2007*). In *Drosophila*, the SMOC orthologue *pentagone (pent)* is expressed in developing wing imaginal discs and has also been shown to inhibit BMP signaling (*Vuilleumier et al., 2010*). However, Pent did not appear to inhibit BMP signaling in the presence of the constitutively active zebrafish BMPR1 receptor, Alk8 (*Vuilleumier et al., 2010*). Structurally, SMOC and Pent are similar, containing an N-terminal follistatin-like domain (FS), followed by two thyroglobulin-like domains separated by a non-homologous domain, and a C-terminal extracellular calcium binding (EC) domain (*Vannahme et al., 2002*; *Vuilleumier et al., 2010*). Although Pent contains an additional EC domain located between the two thyroglobulin-like domains, their phylogenetic conservation would predict similar functions. In *Drosophila*, Pent binds to the cell surface HSPG, Dally, which is required *for Pent to extend the range of BMP signaling during Drosophila* wing patterning (*Vuilleumier et al., 2010*). While SMOC has also been shown to bind to HSPGs (*Klemenčič et al., 2013*), there have been no reports of SMOC promoting BMP signaling at a distance from its source. Indeed, the dual function of SMOC/Pent as a BMP inhibitor and an expander of BMP signaling have not been reconciled. Here, we present the first evidence showing how different domains within SMOC function either to inhibit BMP signaling locally or expand its range of effect.

## Results and discussion

### *Drosophila* Pent can inhibit BMP signaling in *Xenopus* downstream of the BMP receptor

To support the applicability of functional information from Pent to the vertebrate SMOC (*Figure 1A*), we first confirmed that a *pent* cDNA construct was biologically active in *Drosophila*. As

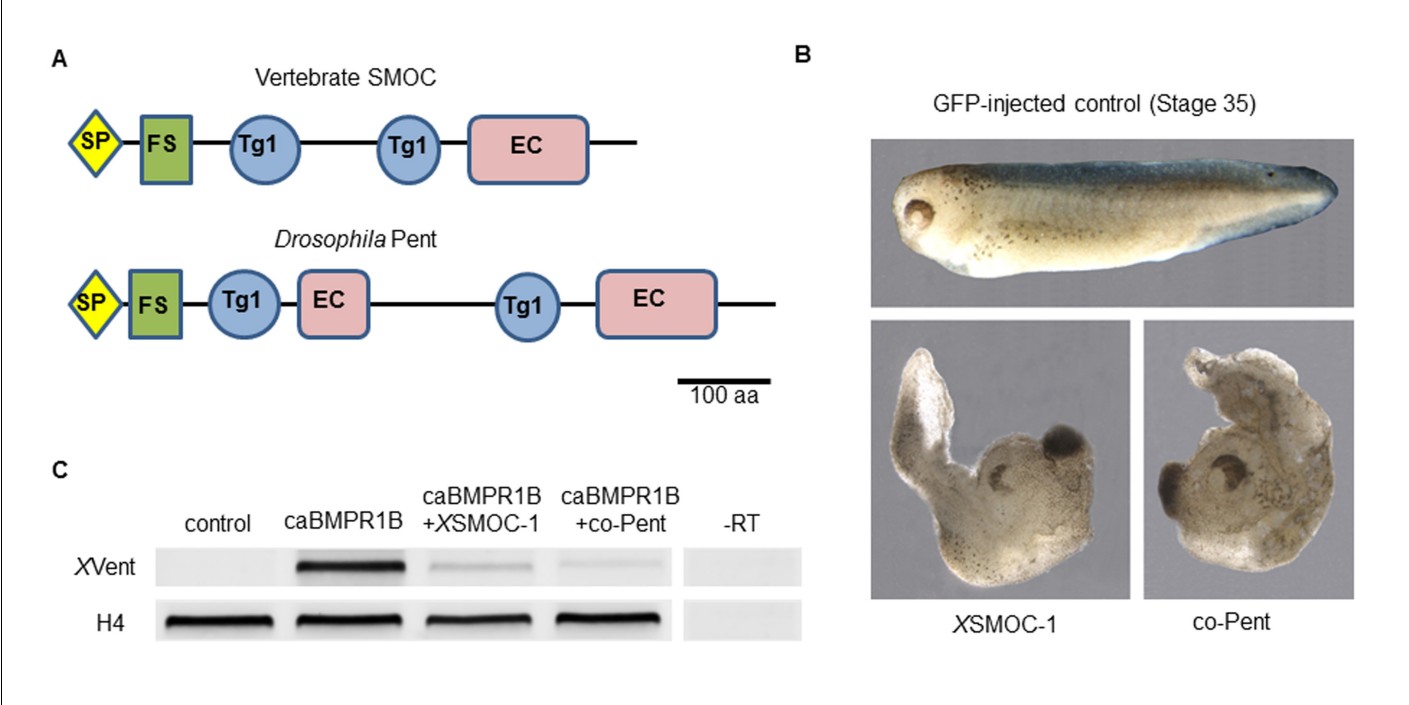

**Figure 1.** *Xenopus* SMOC-1 and *Drosophila pent* are orthologues that inhibit BMP signaling downstream of the BMP receptor. (**A**) Schematic representation of vertebrate SMOC and *Drosophila* Pent: SP- signal peptide, FS – Follistatin-like domain, Tg1 – Thyroglobulin type I-like domain, EC-Extracellular calcium binding domain (**B**) Dorsalized phenotypes of stage 35 *Xenopus* embryos following overexpression of mRNAs for *X*SMOC-1 or codon-optimized *pent* (*co-pent*): *Xenopus* embryos were injected bilaterally at the two-cell stage with 200 pg of mRNA for either GFP (control), *X*SMOC-1, or *co-pent*. The exaggerated dorsal/anterior structures and diminished posterior structures observed following *X*SMOC-1 or *co-pent* overexpression were observed in 95% of embryos in four independent experiments (n = 130). (**C**) RT-PCR analysis of animal cap (AC) explants removed from stage 8/9 embryos injected bilaterally at the two-cell stage with either 450 pg of GFP (Control), 150 pg of constitutively active BMP receptor IB (caBMPR-IB) plus GFP (300 pg), or 150 pg of caBMPR-1B plus *X*SMOC-1 or co-*pent* (300 pg) mRNAs. The AC explants were incubated until stage 20 before RNA extraction. Induction of the BMP signaling target gene, *X*Vent, by caBMPR-1B was blocked by co-expression of either *X*SMOC-1 or co-*pent*. –RT control, without reverse transcriptase.

The following figure supplements are available for figure 1:

**Figure supplement 1.** *Pent* constructs are biologically active in *Drosophila*, but *X*SMOC-1 protein is not detected in *Drosophila* following mRNA overexpression.

**Figure supplement 2.** Nucleotide alignment of *pentagone* and codon optimized *pentagone* (*co-pentagone*).

reported previously (*Vuilleumier et al., 2010*), compared to controls, flies homozygous for *pent* mutations display a characteristic truncation of the L5 longitudinal vein of the adult wing (*Figure 1—figure supplement 1*). When *pent* was expressed in its normal location during wing development, using the Gal4/UAS system, the mutant phenotype was rescued completely, demonstrating that the *pent* construct had full biological activity (*Figure 1—figure supplement 1*). Initial injections of *pent* mRNA into *Xenopus* embryos produced no apparent effects (not shown); however overexpression of a synthetic *pent* mRNA (*co-pent*) optimized for codon usage and translation efficiency in *Xenopus* (*Villegas and Kropinski, 2008*) (*Figure 1—figure supplement 2*) produced a dorsalized phenotype indistinguishable from that observed following overexpression of *Xenopus* SMOC-1 (*X*SMOC-1) (*Figure 1B*). The ability of Pent to inhibit BMP signaling downstream of the BMP receptor was analyzed in *Xenopus* ectodermal explants (animal caps) following co-injection of mRNAs for *co-pent* and constitutively active BMP receptor1B (caBMPR1B); the caBMPR1B, containing an intracellular activating mutation (Q203D), promotes phosphorylation of Smad 1/5/8 and subsequent BMP signaling events independent of ligand binding (*Zou et al., 1997*). As expected, *X*Vent, a direct downstream

target of BMP signaling (*Gawantka et al., 1995*), was strongly expressed in animal caps from embryos injected with caBMPR1B alone (*Figure 1C*). In contrast, *XVent* was markedly reduced in caps from embryos co-injected with caBMPR1B and either *co-pent* or *X*SMOC-1 (*Figure 1C*); these results suggested that, as we had shown previously for SMOC (*Thomas et al., 2009*), Pent can inhibit BMP signaling downstream of the BMPR1B receptor. It is unclear why, in a previous report (*Vuilleumier et al., 2010*), Pent was able to rescue ventralization caused by overexpression of Bmp2b in *Zebrafish*, but not following overexpression of the *Zebrafish* caBMPR1, Alk8. As both BMPR1B (Alk6) and Alk8 are type I BMP receptors that activate Smad 1/5/8, a possible reason could be the amounts of mRNA injected and/or the amounts of protein produced. In our previous study (*Thomas et al., 2009*), SMOC transfected NIH3T3 cells were able to inhibit BMP signaling in the presence of excess BMP2 (100 ng/ml); where the limiting factor would be the number of BMP receptors on the cells. Conversely, in order for SMOC/Pent to inhibit BMP signaling in the presence of caBMPR1B, SMOC/*pent* needed to be overexpressed at a 2:1 ratio (*Figure 1C*); SMOC was not able to inhibit BMP signaling in the presence of an excess of caBMPR1B (not shown).

Having established that Pent can function as a BMP antagonist in *Xenopus* assays, we wanted to determine whether SMOC could function as an expander of BMP signaling in *Drosophila*. However, attempts to express *X*SMOC-1 in *Drosophila* using a synthetic construct optimized for codon usage in *Drosophila* were unsuccessful; immunoblot analysis of two *Drosophila* lines generated to overexpress *X*SMOC-1 demonstrated that *X*SMOC-1 was below the level of detection (5 ng) for the assay (*Figure 1—figure supplement 1*), suggesting that despite codon optimization, *X*SMOC-1 was not translated in amounts sufficient to be effective.

## SMOC expressed in bacteria and refolded is biologically active in *Xenopus* and in mammalian cell lines

As SMOC and Pent are structurally similar, SMOC may function as an expander of BMP signaling in vertebrates. To address this possibility we developed assays using SMOC expressed in bacteria and refolded together with two SMOC deletion mutant constructs; *X*SMOC-1ΔEC lacking the EC domain, and *X*SMOC-1EC containing the EC domain only. The EC domain was of interest as hSMOC-1 binds to heparan sulphate proteoglycans (HSPGs) via the EC domain (*Klemenčič et al., 2013*) and the expander function of Pent is associated with binding to the cell surface-associated HSPG, Dally (*Vuilleumier et al., 2010*). For expression of *X*SMOC-1 in bacteria, the predicted signal peptide (2-24) was omitted and a C-terminal hexahistidine-tag added. When first expressed, two predominant induced proteins were observed on SDS-PAGE (*Figure 2A*); one migrating at 49 kDa, the other at approximately 24 kDa. Protein sequencing revealed the 49 kDa protein to be *X*SMOC-1, whereas the 24 kDa protein was a partial *X*SMOC-1 sequence beginning at V235. The base sequence (GTG) was consistent with an alternative start codon (*Villegas and Kropinski, 2008*); when changed to GTA by site-directed mutagenesis only the expected 49 kDa product was produced (*Figure 2A*). All subsequent *X*SMOC-1 and *X*SMOC-1ΔEC constructs contained GTA at V235.

Initial *X*SMOC-1 refolding studies were conducted using the protocol described previously, where calcium is absent, and produces hSMOC-1 that is monomeric (*Novinec et al., 2008*). Analysis by S-200 size-exclusion chromatography (SEC) showed a mixture of poorly separated peaks, one of which had a calculated molecular weight of 45.5 kDa, approximate to the predicted 49.6 kDa of monomeric *X*SMOC-1 (*Figure 2B*). However, when tested in the *Xenopus* animal cap assay the protein was inactive (not shown). As occupancy of the calcium binding sites may be necessary for biological activity, we modified the refolding buffer to include 2 mM $CaCl_2$. With this change, in addition to poorly-separated higher molecular weight material, both *X*SMOC-1 (*Figure 2C*) and *X*SMOC-1ΔEC (*Figure 2D*) migrated as single, symmetrical peaks. Their calculated molecular weights of 95.4 kDa and 53 kDa, respectively, were approximately twice the predicted monomeric sizes (49.6 kDa and 32.3 kDa). This suggested that *X*SMOC-1 and *X*SMOC-ΔEC refolded in the presence of calcium form dimers. Furthermore, hSMOC-1, shown previously to elute as a monomer (*Novinec et al., 2008*), also migrated as an apparent dimer under these conditions with a calculated molecular weight of 90.3 kDa (*Figure 2C*); no peak was observed at the expected monomeric size of 46.6 kDa. In contrast, *X*SMOC-1EC eluted at the calculated molecular weight of 23.7 kDa (*Figure 2E*), consistent with that predicted for a monomer (18.4 kDa). Dimeric *X*SMOC-1 could not be dissociated by chelation of $Ca^{++}$ ions and continued to elute as a dimer following dialysis in the presence of 10.5 mM EDTA (*Figure 2—figure supplement 1*). Indeed, a dimer was still observed in

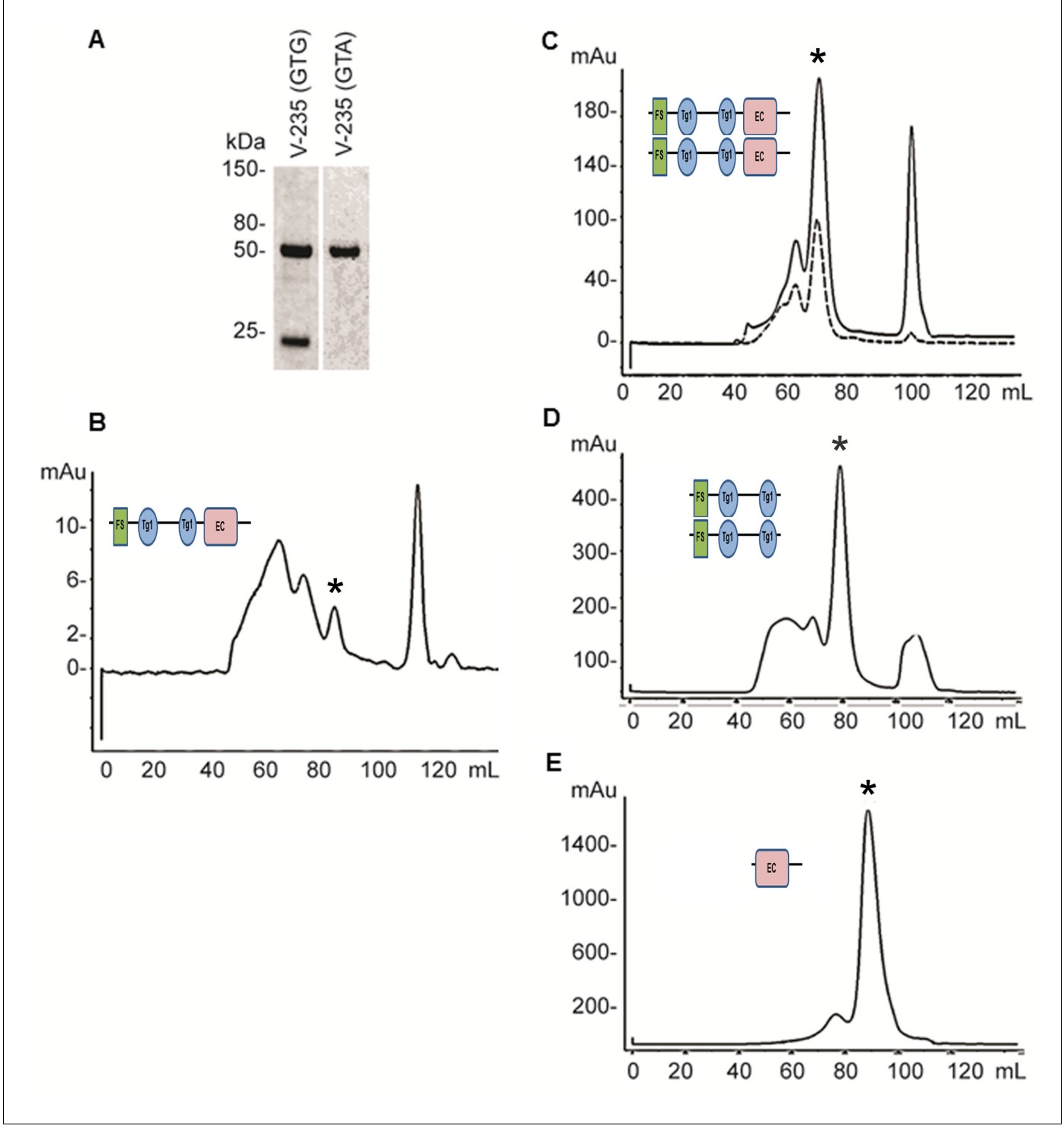

**Figure 2.** Expression and refolding of recombinant mature *X*SMOC-1, *X*SMOC-1ΔEC, and *X*SMOC-1EC. (**A**) Coomassie stained SDS-PAGE of wild type (V235-GTG) and silent mutant (V235-GTA) recombinant mature *X*SMOC-1 following size exclusion chromatography (SEC). The product migrating at 24 kDa is a partial *X*SMOC-1 sequence beginning at the cryptic start site encoded by GTG at V235. (**B–E**) Solid lines: SEC profiles obtained following refolding of *X*SMOC-1 (**B, C**), *X*SMOC-1ΔEC (**D**) and *X*SMOC-1 EC (**E**) either in the absence (**B**) or presence of 2 mM Calcium Chloride (**C–E**). Dashed line: SEC profile (**C**) obtained for human SMOC-1 refolded in the presence of calcium. Asterisk symbols indicate the peaks corresponding to each schematic diagram.

The following figure supplement is available for figure 2:

*Figure 2 continued on next page*

*Figure 2 continued*

**Figure supplement 1.** Dimeric *X*SMOC-1 is not dissociated by chelation or reduction.

the presence of 50 mM EDTA and 1 mM nitriloacetic acid (not shown). Analysis by SDS-PAGE under non-reducing and reducing conditions demonstrated that dimeric *X*SMOC-1 and *X*SMOC-ΔEC were not formed through disulfide linkages (*Figure 2—figure supplement 1*).

## BMP inhibition and neural induction by SMOC does not require the EC domain

The biological activity of the SMOC proteins was assessed using the *Xenopus* animal cap explant assay in which overexpression of *X*SMOC-1 mRNA was shown previously to induce anterior neural markers (*Thomas et al., 2009*). *X*SMOC-1, *X*SMOC-1ΔEC, or *X*SMOC-1EC proteins were incubated with stage 9 (*Nieuwkoop and Faber, 1994*) late blastula animal caps, at equimolar concentrations, until wild type embryos reached the late neurula stage (stage 20–21). RT-PCR demonstrated the induction of anterior neural markers in the presence of *X*SMOC-1 and *X*SMOC-1ΔEC, but not *X*SMOC-1EC (*Figure 3A*); hSMOC-1 was also effective in this assay (not shown). For these studies, the optimal concentration of SMOC was found to be 100 µg/ml, which appears relatively high. However, SMOC is a matricellular protein and though it is difficult to estimate the effective concentration in vivo, its affinity for HSPGs suggests that its diffusion will be restricted unless HSPG sites are saturated. Consequently, it may remain concentrated near the site of secretion and thus achieve high levels locally. Temporal analyses showed that a two-hour pulse of *X*SMOC-1 or *X*SMOC-1ΔEC protein was sufficient to commit the naïve ectoderm of the *Xenopus* animal cap to an anterior neural fate sixteen hours later (*Figure 3B*); a one-hour pulse was not. This suggests that, following a two hour exposure to SMOC, a duration of exposure (*Rogers and Schier, 2011*) is reached whereby sufficient changes in gene transcription occur in SMOC-responsive cells to convert their fate from epidermal to neural. While it is well established that inhibition of endogenous BMP activity in *Xenopus* ectodermal explants promotes a neural fate (*Vonica and Brivanlou, 2006*) and a number of genes have been implicated in neural fate specification (*Kishi et al., 2000*; *Ueno et al., 2008*; *Milet et al., 2013*; *Green and Vetter, 2011*; *Gammill and Sive, 2001*), the sequence of events resulting in commitment to the neural lineage is not known. It should now be feasible to design an unbiased genome-wide screen to identify the early transcriptional changes, following a two hour exposure to SMOC, that initiate neural differentiation.

The ability of the SMOC proteins to inhibit BMP signaling was assessed in mammalian cell lines. Serum-starved NIH-3T3 and HEK-293 cells were incubated with either *X*SMOC-1 (100 µg/ml) or equimolar amounts of *X*SMOC-1ΔEC or *X*SMOC-1EC for thirty minutes, followed by the addition of BMP2 (50 ng/ml) for an additional thirty minutes. As expected, BMP2 treatment alone caused phosphorylation of Smad 1/5/8 (*Figure 4A,B*). This was blocked in the presence of *X*SMOC-1 and *X*SMOC-1ΔEC, whereas *X*SMOC-1EC significantly enhanced BMP2-mediated Smad phosphorylation (*Figure 4A,B*). Potentiation of Smad phosphorylation by the EC domain was not due to an additive effect, as the addition of *X*SMOC-1EC alone to NIH-3T3 cells did not result in Smad1/5/8 phosphorylation (*Figure 4C*).

Having established that the SMOC EC domain is not required for the inhibition of BMP signaling we designed deletion constructs for use in mRNA overexpression studies to determine which domain(s) of SMOC are required for BMP inhibition; *X*SMOC-1ΔFS contained a 49 amino acid deletion of the N-terminal FS-like domain (ΔQ43 to A91); *X*SMOC-1ΔTg1 contained only the follistatin-like domain and EC domain (ΔK95 to S304); *X*SMOC-1ΔFSΔEC (referred to as *X*SMOC-1Tg1) contained deletions of the FS-like and EC domains (ΔQ43 to A91 and ΔN310 to end), leaving only the two Tg1-like domains. The effect on BMP inhibition was analyzed in *Xenopus* embryos following overexpression of mRNAs for each deletion construct and caBMPRIB. Analysis of animal caps following co-injection of *Xenopus* embryos with caBMPRIB and *X*SMOC-1, *X*SMOC-1ΔFS, or *X*SMOC-1ΔEC showed inhibition of BMP signaling, indicated by the suppression of *X*Vent expression (*Figure 4D*). Whereas these data suggest that the FS and EC domains are not required for BMP inhibition overexpression of *X*SMOC-1Tg1, containing the Tg1-like domains only, did not inhibit BMP

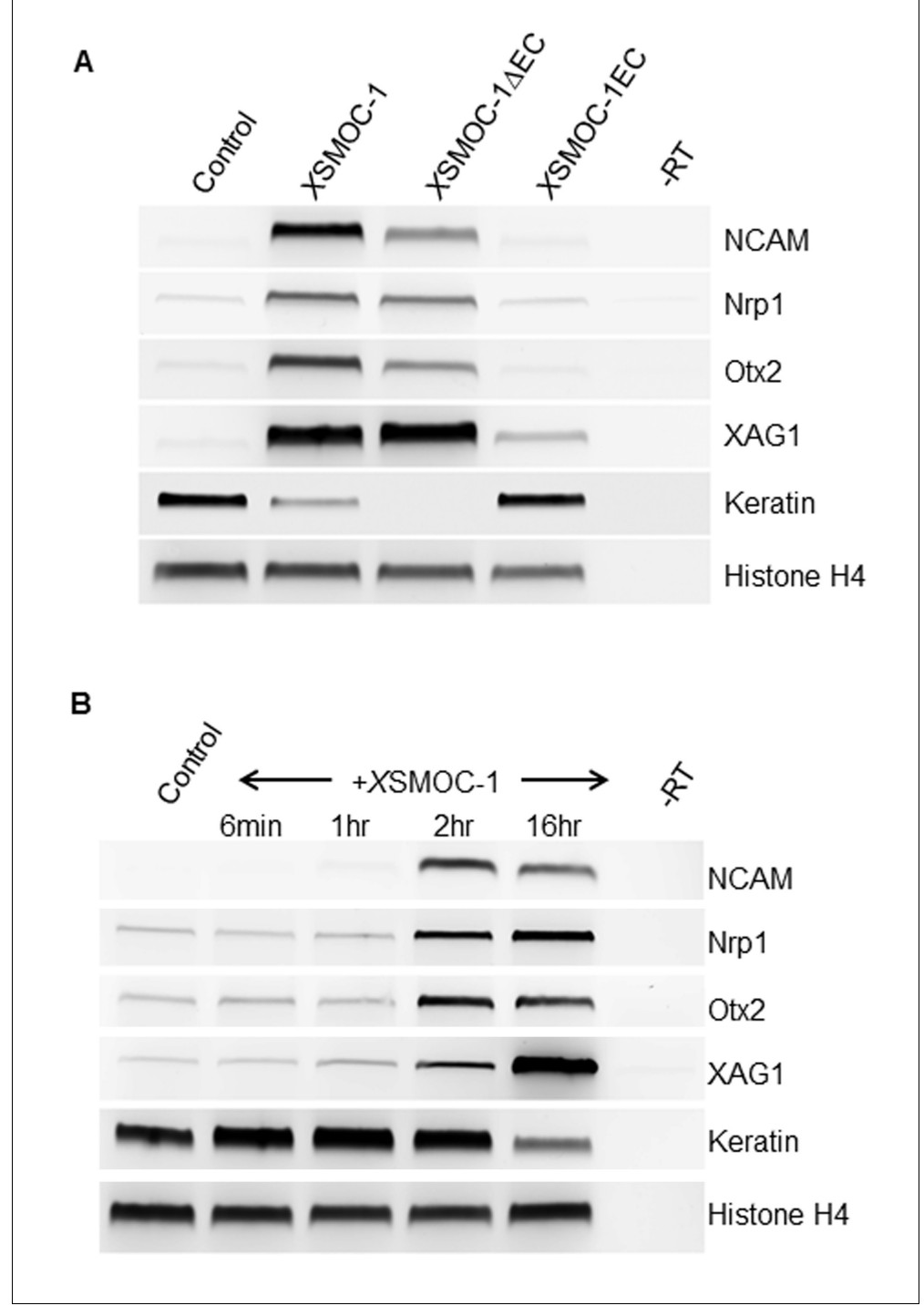

**Figure 3.** *X*SMOC-1 and *X*SMOC-1ΔEC, but not *X*SMOC-1EC convert the fate of naïve *Xenopus* ectoderm explants (animal caps) to anterior neural tissue within two hours. (A) RT-PCR analysis of animal caps removed at stage 8/9 and incubated in 0.7X MMR/0.1% BSA (control) containing equimolar amounts of *X*SMOC-1 (100 μg/ml), *X*SMOC-1ΔEC (75 μg/ml) or *X*SMOC-1EC (50 μg/ml) until sibling embryos reached the late neurula stage (20); anterior neural markers (N-CAM, Nrp-1, Otx2, Xag-1) were induced by both *X*SMOC-1 and *X*SMOC-1ΔEC, but not by *X*SMOC-1EC. Expression of the ectodermal marker Keratin was suppressed by both *X*SMOC-1 and *X*SMOC-1ΔEC, but not by *X*SMOC-1EC. (B) Animal caps removed at stage 8/9 were incubated in 0.7X MMR/0.1% BSA (control) in the presence of *X*SMOC-1 (100 μg/ml) for six minutes, one hour, or two hours before replacing with 0.7X MMR/0.1% BSA and incubating until sibling embryos reached stage 20. RT-PCR analysis shows that a two hour exposure to *X*SMOC-1was sufficient to induce the naïve ectoderm to express anterior neural markers 16 hr

*Figure 3 continued on next page*

*Figure 3 continued*

post-pulse; a one hour exposure was not. The continual presence of *X*SMOC-1 (16 hr) was used as a positive control.

signaling in this assay (*Figure 4D*); deletion of the Tg1-like domains (*X*SMOC-1ΔTg1) produced a similar result. While we believe the Tg1-like domains to be important for BMP inhibition, by process of elimination, the effects of large deletions on protein folding are difficult to predict; in experiments of this type, where protein function is lost, improper folding is common (*Valastyan and Lindquist, 2014*). Consequently, we consider misfolding of the protein produced by *X*SMOC-1ΔFSΔEC as the most likely reason for the inability of this construct to block BMP signaling. Additional evidence for the importance of the Tg1-like domains in SMOC comes from studies of human Ophthalmo-Acromelic (Waardenburg Anophthalmia) Syndrome, an autosomal disorder caused by mutations in SMOC-1 (*Rainger et al., 2011*). The phenotype of anopthalmia, oligodactyly, and joint abnormalities

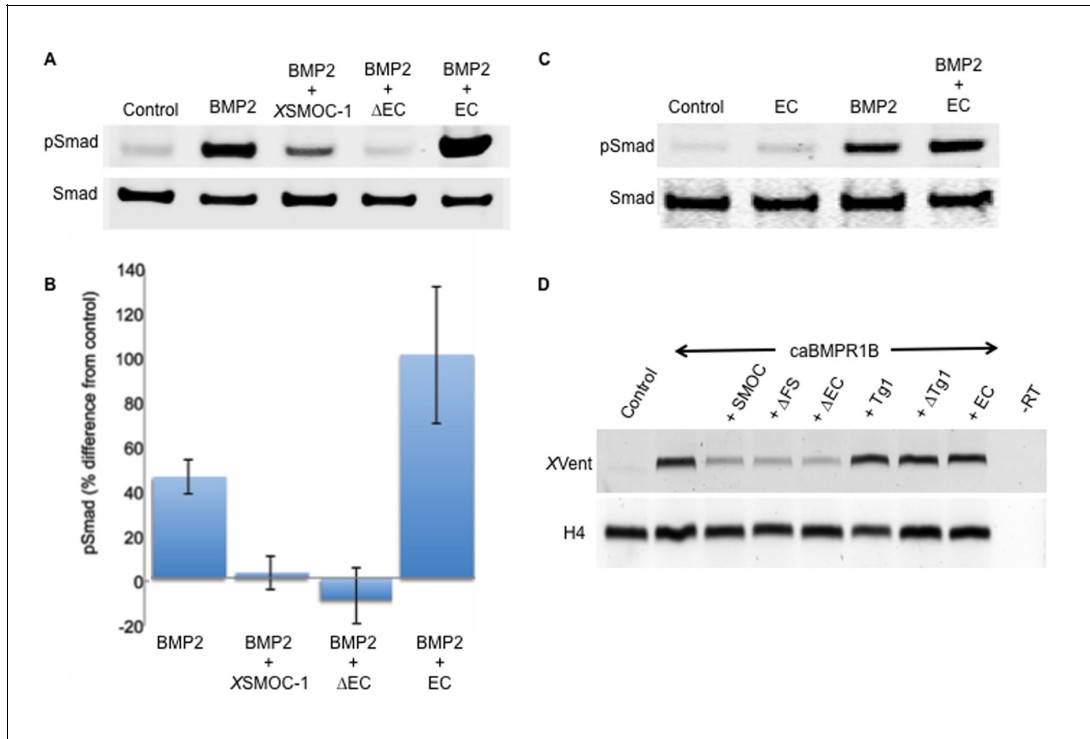

**Figure 4.** The two thyroglobulin-like domains are necessary for BMP inhibition, whereas the EC domain promotes BMP signaling. (**A**) Immunoblot showing phosphorylation of Smad 1/5/8 (pSmad) by BMP2 (50 ng/ml) in HEK-293 cells is inhibited by the addition of *X*SMOC-1 (100 µg/ml) or *X*SMOC-1ΔEC (75 µg/ml), but is enhanced by the addition of *X*SMOC-1EC (50 µg/ml). Total Smad is shown as a loading control. (**B**) Graph showing the relative fluorescence of pSmad 1/5/8 obtained on immunoblots from four separate experiments using both HEK293 and NIH3T3 cells; each treatment is displayed as the percent difference from control. The inhibition of Smad 1/5/8 phosphorylation by *X*SMOC-1 or *X*SMOC-1 ΔEC and the potentiation of BMP signaling by *X*SMOC-1EC are both significant (p=≤0.01). (**C**) Immunoblot of HEK293 cell extracts showing that *X*SMOC-1 EC alone does not promote Smad phosphorylation. (**D**) RT-PCR analysis of *Xenopus* animal cap (AC) explants removed at stage nine from embryos injected bilaterally at the two-cell stage with mRNAs for GFP (Control), caBMPR1B (120 pg), or caBMPR1B and equimolar amounts of *X*SMOC-1(600 pg), *X*SMOC-1 ΔFS (540 pg), *X*SMOC-1 ΔEC (420 pg), *X*SMOC-1 Tg1 (330 pg), *X*SMOC-1 ΔTg1 (360 pg) or*X*SMOC-1 EC (240 pg). The AC explants were incubated until stage 20 (late neurula) before RNA extraction and analysis. The induction of the direct BMP signaling target gene, *X*Vent, by caBMPR1B was blocked by co-expression with *X*SMOC-1, *X*SMOC-1 ΔFS, or *X*SMOC-1 ΔEC, but not by *X*SMOC-1 Tg1, *X*SMOC-1 ΔTg1, or *X*SMOC-1 EC. H4: Histone loading control, –RT: Negative control.

The following source data is available for figure 4:

**Source data 1.** Source data file for generating *Figure 4B*.

was found to be the same in patients with nonsense or frameshift mutations and those with missense mutations in the second Tg1-like domain (*Rainger et al., 2011*). As the nonsense and frameshift mutations were predicted to result in a complete loss of SMOC-1 function, the two pedigrees harboring two different single amino acid missense mutations in the Tg1-like domain suggests this domain is indeed essential for SMOC-1 function. Alternative approaches will be required to elucidate the exact role of the Tg1-like domains in BMP inhibition. Many proteins contain Tg1-like domains, including thyroglobulin, insulin-like growth factor binding proteins (IGFBPs) 1–6, the proteoglycan testican, and the basement membrane associated protein nidogen/entactin (*Novinec et al., 2006*). However, there have been no reports of any of these proteins inhibiting BMP signaling.

## The SMOC EC domain can expand the range of BMP signaling in vitro by competitive binding to HSPGs

The potentiation of BMP signaling by the EC domain was examined further by investigating the relative affinities of XSMOC-1EC and BMP2 for each other and for HSPGs. The binding of SMOC/Pent and BMP2/4 to HSPGs is known, as evidenced by the co-purification of SMOC and BMPs following heparin affinity chromatography of bovine cartilage extracts (*Chang et al., 1994*), and the binding of BMP2/4 and the EC domain of hSMOC/Pent to heparin/HSPGs (*Vuilleumier et al., 2010*; *Klemenčič et al., 2013*; *Ruppert et al., 1996*). In addition, the basic amino acid-rich putative heparin-binding region identified within the EC domain of SMOC (*Klemenčič et al., 2013*) is highly conserved in Pent (*Figure 5—figure supplement 1*). Using the Protein Homology/analogY Recognition Engine (PHYRE), an unsupervised homology model for XSMOC-1EC was constructed based on the structure of the EC domain of the related family member BM-40 (*Hohenester et al., 1996*). XSMOC-1EC aligned well with the BM-40-EC model (*Figure 5—figure supplement 1*) and the electrostatic surface potential map predicted an area of positive charge similar to that reported in the EC domain of hSMOC1 (*Klemenčič et al., 2013*) (*Figure 5—figure supplement 1*). As monomeric hSMOC-1, refolded in the absence of calcium, was used in previous heparin-binding studies (*Klemenčič et al., 2013*), we first determined whether dimeric XSMOC-1 can bind heparin. XSMOC-1 and XSMOC-1EC bound to heparin Sepharose in the presence of 0.5M NaCl, whereas XSMOC-1ΔEC did not (*Figure 5A*), confirming the EC domain to be the site of HSPG binding. Comparison of the heparin-binding affinities of XSMOC-1EC and BMP2 showed a striking similarity, with both eluting between 0.65M and 0.7M NaCl (*Figure 5B*). The possibility that XSMOC-1 and BMP2 bind to each other was discounted (*Figure 5C*); when BMP2 was incubated with XSMOC-1 or XSMOC-1 EC, pre-bound to heparin-Sepharose, BMP2 was only present in the unbound fraction (*Figure 5C*; lanes 2 and 5) and did not co-elute with XSMOC-1 or XSMOC-1 EC (*Figure 5C*; lanes 3 and 6). The lack of interaction of SMOC with BMP2 agrees with an analogous finding observed for Pent and the *Drosophila* BMP, decapentaplegic (Dpp) (*Vuilleumier et al., 2010*). This, combined with SMOC and BMP2 having similar heparin-binding affinities, suggests that XSMOC-1EC could compete with BMPs for HSPG binding on HEK293 cells and thereby increase BMP bioavailability.

We designed an in vitro assay to test the hypothesis that SMOC can expand the range of BMP signaling by competing with BMP2/4 for HSPG-binding. BMP4-soaked beads represented a cellular source of BMPs and agarose gels (0.7%) containing heparan sulfate (HS) (10 µg/ml) represented an extracellular matrix (ECM) capable of binding SMOC and BMPs. Chamber slides containing BMP4-soaked beads embedded in the agarose/HS/XSMOC-1EC matrices were seeded with the stable reporter cell line C33A-2D2-09, harboring luciferase under the control of a BMP response element (BRE). After 24 hr, immunohistochemical analysis of cells in fields of view adjacent to the matrices showed many luciferase positive cells (59%) adjacent to the agarose-only gel (*Figure 6A,D*), whereas when HS was present only a few (8%) were detected (*Figure 6B,E*). In contrast, when the matrix contained both HS and XSMOC-1EC, the number of BMP-responsive cells (64%) was similar to that observed in the absence of HS (*Figure 6C,F*). The result showed that only a relatively small amount of BMP4 diffused through the HS-containing agarose gel, activating luciferase expression in only a few BMP-responsive cells. When XSMOC-1EC was present, sufficient BMP4 diffused through the HS-containing matrix to activate a greater number of cells. The assay demonstrated that binding of BMPs to HSPGs restricts their range of effect and that BMP diffusion can be enhanced by the binding of SMOC to HSPGs, effectively expanding the range of effect.

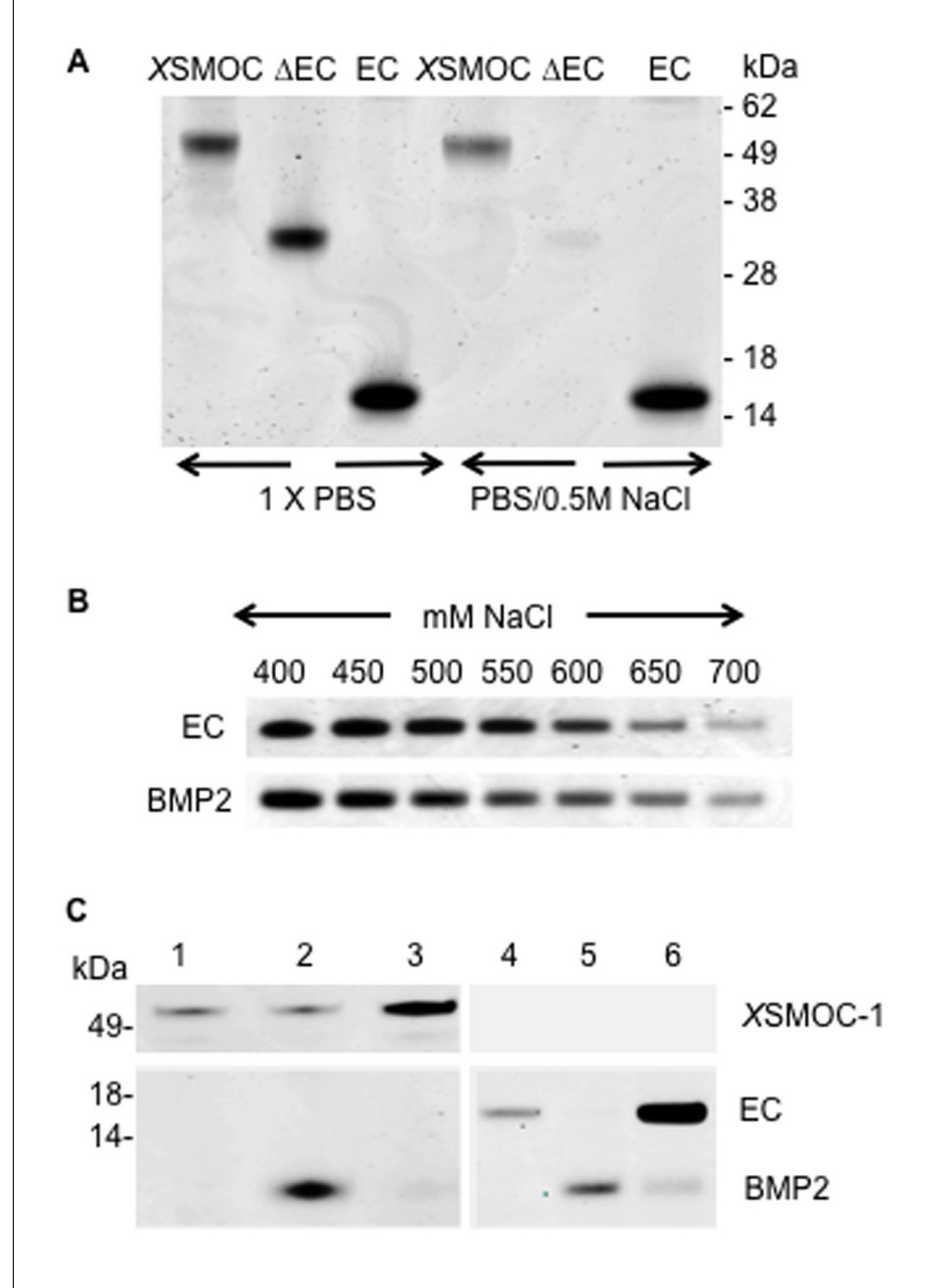

**Figure 5.** XSMOC-1EC and BMP2 have similar binding affinities for heparin sepharose (HS), but do not bind to each other. SDS-PAGE analysis (Coomassie staining) of HS elution profiles showing (**A**) binding of XSMOC-1, XSMOC-1ΔEC, and XSMOC-1EC in PBS or PBS/0.5M NaCl; binding of XSMOC-1 to HS requires the EC domain. (**B**) XSMOC-1EC and mature BMP2 (A284-R396) in a NaCl gradient (400–700 mM) have equivalent HS binding affinities. (**C**) BMP2 does not bind directly to XSMOC-1 (lanes 1–3) or XSMOC-1EC (lanes 4–6) at physiological ionic strength (PBS); BMP2 (4 μg) incubated with HS (0.3 μl) saturated with XSMOC-1 or XSMOC-1EC (6 μg), did not co-elute with XSMOC-1 or XSMOC-1EC (lanes 3 and 6) and was only present in the unbound fraction (lanes 2 and 5). Saturation of HS by XSMOC-1 and XSMOC-1EC was confirmed by their presence in the unbound fractions (lanes 1 and 4) prior to incubation with BMP2.

The following figure supplement is available for figure 5:

**Figure supplement 1.** The EC domains of SMOC and Pent are conserved and share structural homology to BM-40.

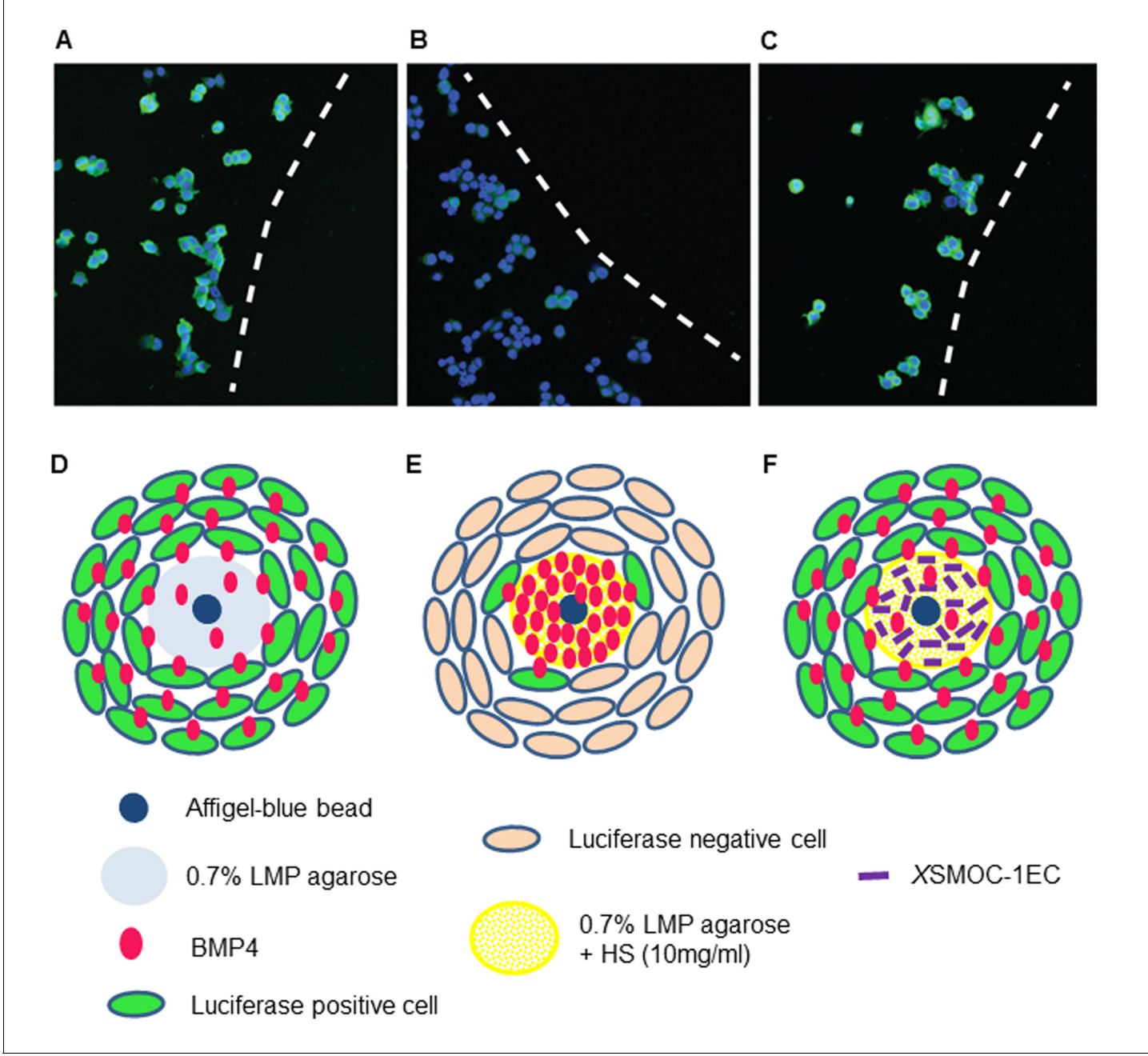

**Figure 6.** Immunofluorescence assay demonstrating that *X*SMOC-1 can promote BMP signaling at a distance from its source by competitive binding to HSPGs. BMP4-soaked beads (100 μm) were implanted into 0.5 μl drops of 0.7% low melting point (LMP) agarose (**A, D**), LMP agarose containing 10 μg/ml heparan sulfate (**B, E**), or LMP agarose containing heparan sulfate (10 μg/ml) and 100 μg/ml *X*SMOC-1EC (**C, F**) on 8-well chamber slides. C33A-2D2-09 cells, harboring luciferase under the control of a BMP response element (BRE) were seeded at $2 \times 10^4$ cells/well and incubated for 48 hr in serum-free medium. Luciferase immunofluorescence (green) indicates cells positive for BMP signaling. Cell nuclei (blue) were stained with DAPI. Dashed lines in A-C indicate the boundaries between C33A-2D2-09 cells and the agarose drops. Analysis of four fields of view in three separate experiments demonstrated the number of luciferase-positive cells to be lower (16% ± 8%) adjacent to matrices containing HS alone (**B, E**) compared to those containing both HS and *X*SMOC-1EC (69% ± 26%) (**C, F**). Representative fields are shown.

The following source data is available for figure 6:

**Source data 1.** Percentage of luciferase positive cells per field of view.

## SMOC can potentiate BMP signaling at a distance from its source in vivo

We designed an assay to assess the ability of SMOC to affect the range of BMP signaling in vivo using *Xenopus* ectodermal conjugates, from stage 9 to 9.5 *Xenopus* embryos, in which endogenous BMP signaling is absent (*Faure et al., 2000*). Animal caps, removed at stage nine from embryos co-injected with mCherry and/or SMOC mRNAs, were grafted onto the animal poles of BMP2 mRNA-injected embryos from which the caps had been removed (*Figure 7A*). The conjugates were allowed to heal for two hours before the ectoderm from the entire animal half of the chimeric embryos was removed for immunostaining for phospho-Smad 1/5/8 (*Figure 7B–F*). In non-injected controls, no pSmad signal was detected either in the host tissue or the mCherry mRNA-injected donor tissue (*Figure 7B*). When host embryos were injected with a high concentration of BMP2 mRNA (300 pg), pSmad signaling was observed throughout the host and donor graft tissue (*Figure 7C*). Following a series of titration experiments we found that injection of 30 pg BMP2 mRNA at the two cell stage (15 pg into each blastomere) was sufficient to induce BMP signaling in the host tissue, while pSmad was observed only at a few cell diameters into the donor graft (*Figure 7D*). This amount of BMP2 mRNA was used in subsequent experiments to test whether SMOC can extend the range of BMP

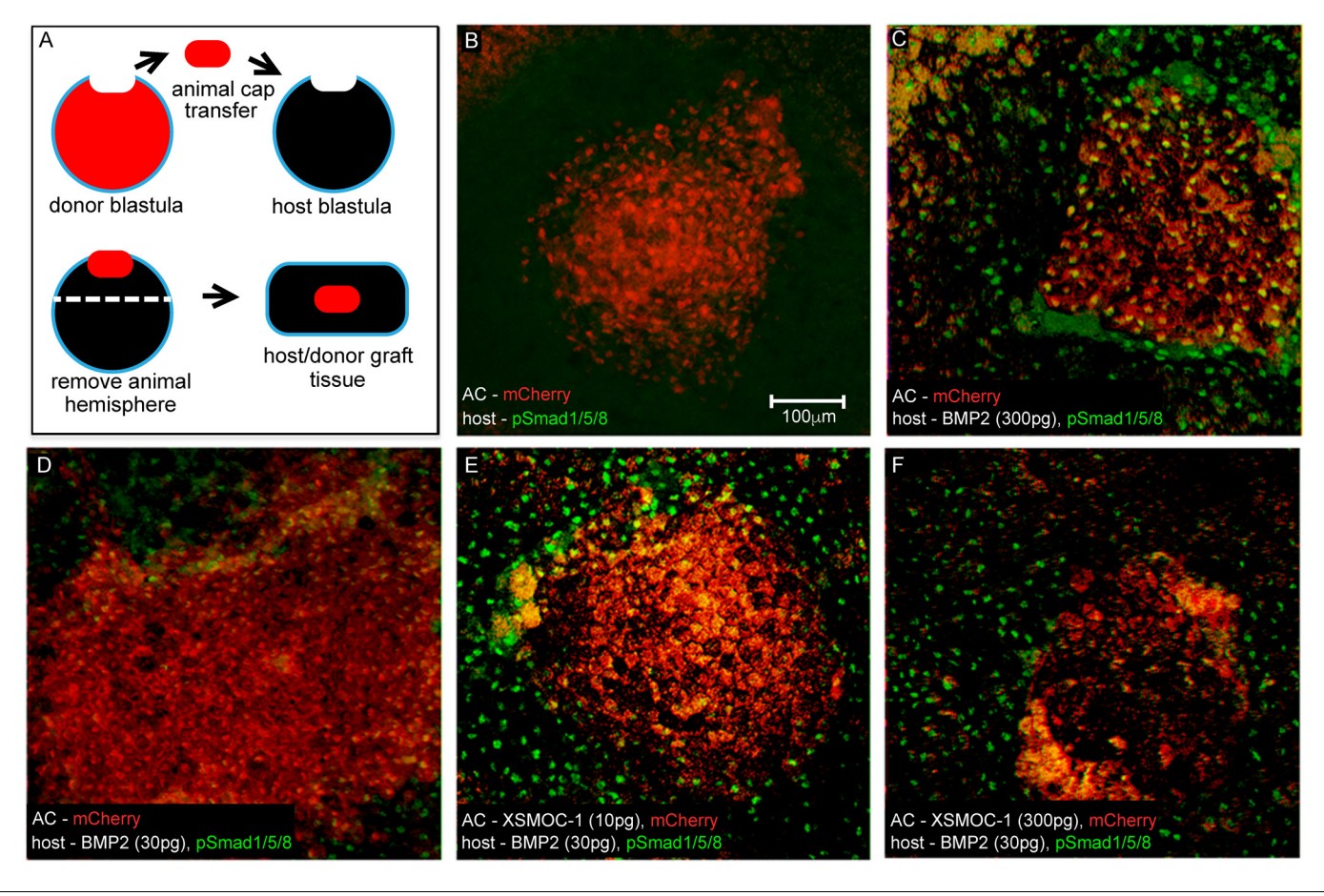

**Figure 7.** In vivo assay demonstrating that *XSMOC-1* can expand the range of BMP2 signaling. (**A**) Schematic diagram of the host/donor animal cap (AC) transfer assay. (**B–F**) Donor AC grafts expressing mCherry (red) and host/donor immunofluorescent nuclear staining of pSmad 1/5/8 (green). (**B**) Control host + mCherry mRNA (200 pg)-injected donor AC (mCherry AC); endogenous pSmad is not detectable, (**C**) BMP2 mRNA (300 pg)-injected host + mCherry AC; pSmad is detected throughout the host tissue and donor AC, (**D**) BMP2 mRNA (30 pg)-injected host (BMP2-30 pg host) + mCherry AC; pSmad is detected in the host tissue and at the host tissue/AC boundary, (**E**) BMP2-30 pg host + mCherry/*XSMOC-1* mRNA (10 pg)-injected AC; pSmad is detected in the host tissue and 4–5 cell diameters into the AC (**F**) BMP2-30pg host + mCherry/*XSMOC-1* mRNA (300 pg)-injected AC; pSmad is not detected in the AC and is also absent at the host tissue/AC boundary.

signaling. Next, donor animal caps from embryos injected with different amounts of SMOC mRNA were grafted onto the animal poles of embryos injected with 30 pg BMP2 mRNA. Donor caps from embryos injected with a low amount of SMOC mRNA (10 pg) had an increased number of pSmad positive nuclei compared to non-SMOC injected caps (*Figure 7E*). Conversely, donor caps from embryos injected with a high amount of SMOC mRNA (300 pg) were devoid of pSmad (*Figure 7F*). The results demonstrate that, in an intact tissue, low concentrations of SMOC can promote diffusion of BMP from its source of synthesis, thereby extending its range of effect, whereas high SMOC concentrations inhibit BMP signaling. These data would be consistent with low levels of SMOC binding to HSPGs, but not inhibiting BMP signaling, and high amounts of SMOC both binding HSPGs and inhibiting BMP signaling. Closer examination of the host tissue surrounding the donor grafts expressing the highest level of SMOC showed BMP signaling was both absent in the graft and decreased in the BMP2-expressing ectoderm immediately adjacent to the SMOC expressing graft (*Figure 7F*). This would be consistent with SMOC inhibiting BMP signaling within the graft and also diffusing into the immediate adjacent tissue at levels sufficient to inhibit BMP signaling.

## Model to explain the dual function of pent and SMOC as BMP antagonists and expanders of BMP signaling

In addition to the data we present here, there are substantial in vivo data from previous work that are consistent with SMOC/Pent acting both as a BMP antagonist and by expanding the range of BMP signaling. During *Drosophila* wing development, while *dpp* expression is restricted to a stripe of medial cells in the wing disc, the morphogenetic gradient expands across the anterior/posterior (A/P) axis (*Nellen et al., 1996*). Long-range Dpp signaling in the wing has been shown to involve both *pent* and the cell membrane-anchored HSPG, *dally* (*Vuilleumier et al., 2010*; *Fujise et al., 2003*); the absence of either causes severe contraction of the range of Dpp signaling, and wing patterning defects (*Vuilleumier et al., 2010*). However, the mechanism by which Pent and Dally cooperate to expand the range of Dpp signaling is not clear. It has been speculated that the Pent/Dally interaction may reduce the affinity of Dpp for its receptor, Thickveins, leading to an increase in Dpp diffusion (*Ben-Zvi et al., 2011*). However, the results we obtained from cell culture studies suggest that this is not the case. When *X*SMOC-1EC is added to HEK293 cells at 50 µg/ml, in serum-free media it will bind to the many cell surface HSPG binding sites. Consequently, following the subsequent addition of BMP2, any impairment of BMP2/receptor affinity caused by SMOC/HSPG binding would result in a reduction of BMP signaling. Instead, a potentiation of BMP signaling was observed (*Figure 4A,B*). Based on the expression patterns of *dpp, pent, and dally*, their known biological activities, and the new information presented here, we propose the following model (*Figure 8A–C*).

*Dally*, expressed both medially and laterally in the wing disc (*Fujise et al., 2003*), encodes a cell autonomous membrane-associated protein; *pent*, expressed laterally, and *dpp*, expressed medially, encode secreted proteins able to diffuse away from their source (*Thomas et al., 2009*; *Vuilleumier et al., 2010*). Indeed, we showed previously that SMOC can diffuse across many cell diameters in *Xenopus* animal caps (*Thomas et al., 2009*). At the lateral border, Dpp signaling will be inhibited by a combination of the expression of *pent* (*Figure 7A*) and the BMP repressor, *brinker* (*Minami et al., 1999*). As Pent diffuses across the wing disc it will create a lateral-medial gradient and compete with Dpp for Dally binding. Dpp, diffusing in an opposing medial-lateral gradient, will encounter gradually increasing levels of Pent. At low levels, Pent will occupy some Dally binding sites, promoting further Dpp diffusion; at high levels, Pent will occupy many Dally binding sites and also inhibit Dpp signaling downstream of its receptor (*Figure 8A*). This model is supported by the observations made either in the absence of *pent* (*Vuilleumier et al., 2010*), or the absence of *dally* (*Fujise et al., 2003*), where there is a contraction of the Dpp signaling gradient. In the absence of *pent*, medial-lateral diffusion of Dpp will be restricted by Dally because there is no Pent to compete for Dally binding sites (*Figure 8B*). The resulting high level of medially localized Dpp signaling, indicated by Mad phosphorylation (pMad) (*Vuilleumier et al., 2010*), has been shown to induce expression of the inhibitory Smad, *daughters against dpp* (*dad*), promoting a negative feedback loop to limit the range of Dpp signaling (*Tsuneizumi et al., 1997*; *Ogiso et al., 2011*). In the absence of *dally*, Dpp diffusion will also be restricted as Dally-binding is required to establish both Dpp and Pent gradients (*Vuilleumier et al., 2010*). Medially, *dally* mutant discs show abnormally high levels of pMad (*Fujise et al., 2003*), which would promote a negative feedback loop in Dpp signaling similar to that observed in the absence of *pent* (*Figure 8C*). Our proposed mode of action of Pent in the

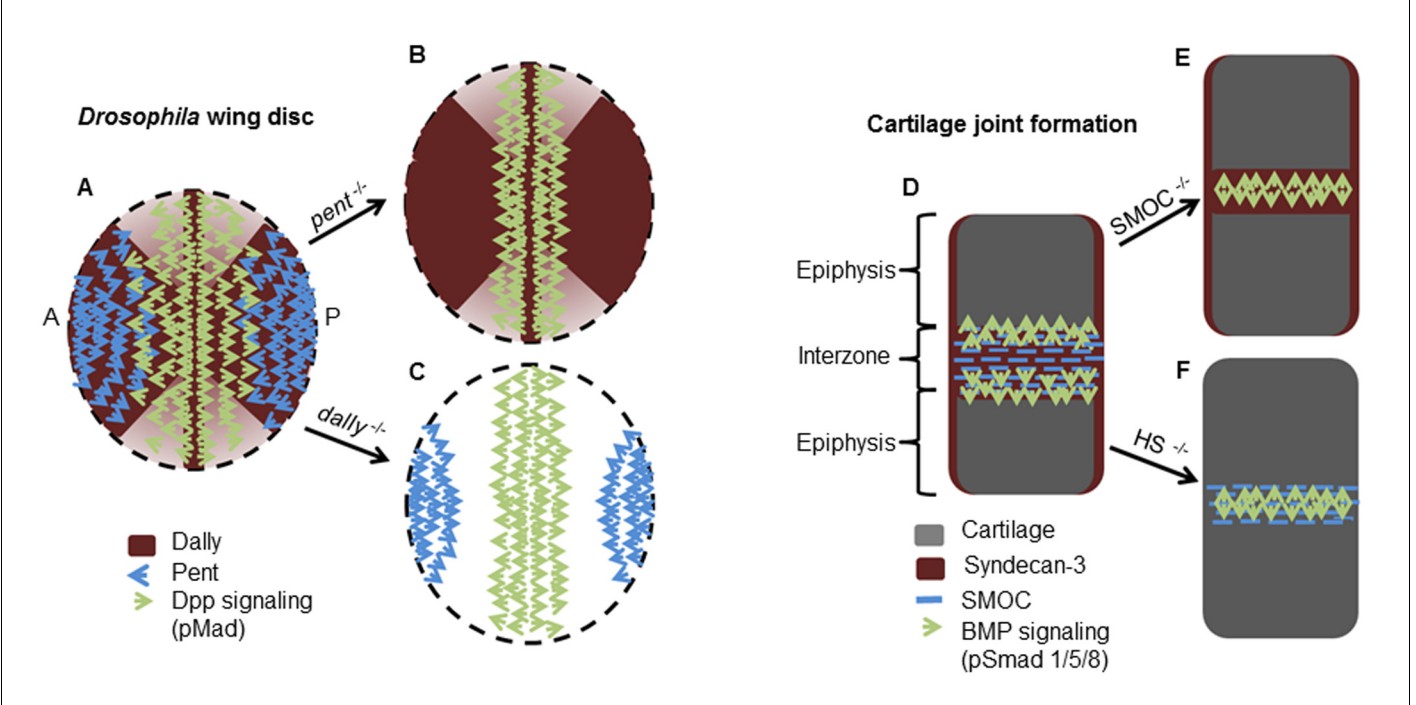

**Figure 8.** Schematic diagrams showing the proposed mode of action of Pent in the *Drosophila* wing disc and SMOC in vertebrate joint interzones. (A) Representations of the expression patterns of *pent* (*Vuilleumier et al., 2010*) *dally* (*Fujise et al., 2003*) and Dpp signaling (pMad) (*Vuilleumier et al., 2010*) in wild type *Drosophila* wing discs, compared with (B) *pent*$^{-/-}$ or (C) *dally*$^{-/-}$ mutant discs. The expression patterns observed in the absence of *pent* or *dally* (*Vuilleumier et al., 2010*) are consistent with Pent/Dally binding being required to expand the range of Dpp signaling. (D) Expression patterns of SMOC (*Okada et al., 2011*; *Rainger et al., 2011*), Syndecan-3 (*Koyama et al., 1995*), and BMP signaling (pSmad) [4949] in a developing vertebrate autopod joint, compared with that predicted for (E) the absence of HSPGs (HS$^{-/-}$), or (F) the absence of SMOC (SMOC$^{-/-}$).

*Drosophila* wing disc is consistent with the role of glypicans in the 'restricted extracellular diffusion' model for establishing a Dpp signaling gradient (*Schwank et al., 2011*); the inclusion of the Pent/glypican interaction provides for further diffusion of Dpp to expand the morphogenetic field. During the review of this manuscript a study was published (*Norman et al., 2016*) showing that binding of Pent to Dally in *Drosophila* induces endocytosis of the complex, effectively reducing the amount of Dally on cell surfaces. Although it remains to be determined whether binding of SMOC to HSPGs also induces endocytosis, this finding is consistent with the model we propose: both competitive binding of SMOC/Pent to HSPGs and removal of HSPG binding sites for BMP/Dpp by endocytosis would promote BMP/Dpp diffusion. This idea is further supported by the observation that removal of the heparin binding site within BMP4 has also been shown to promote the BMP diffusion in *Xenopus* embryos (*Ohkawara et al., 2002*).

The binding of Pent to Dally also modulates Wingless (Wg) signaling in *Drosophila*, where a reduction in Wg signaling is observed in the presence of high levels of Pent (*Norman et al., 2016*). The interaction of Wg with Dally is known to potentiate Wg binding to its receptor Frizzled (*Tsuda et al., 1999*; *Lin and Perrimon, 1999*); therefore, the sequestration of Dally by Pent would be consistent with a reduction in Wg signaling. These data could also provide an explanation for our previous observation that SMOC activates MAPK signaling and inhibits BMP signaling via a mechanism requiring phosphorylation of Smad in the linker region (*Thomas et al., 2009*). Linker Smad phosphorylation at MAPK sites has been shown to be followed by phosphorylation at Glycogen Synthase Kinase 3 (GSK3) sites (*Fuentealba et al., 2007*), resulting in Smad ubiquitination and degradation (*Sapkota et al., 2007*). As the activity of GSK3 is inhibited by Wnt signaling, a reduction in Wnt signaling by Pent/SMOC could potentiate GSK3-mediated phosphorylation of linker Smad.

## The potential function of SMOC during skeletal joint development

The spatial distributions and interactions of *pent*, *dally*, and *dpp* in the *Drosophila* wing disc can also be applied to explain the role of SMOC, HSPGs, and BMPs during development of vertebrate skeletal joints (*Figure 8D–F*). SMOC-1 (*Okada et al., 2011*), the cartilage-specific BMP called growth and differentiation factor 5 (GDF5) (*Storm and Kingsley, 1996*) or cartilage-derived morphogenetic protein-1 (CDMP-1) (*Chang et al., 1994*), BMP2/4 (*Francis-West et al., 1999*), and HSPGs (*Koyama et al., 1995*) are all expressed at the sites of future joint interzones; the absence of any one of these results in dysmorphogenesis of autopod joints (*Okada et al., 2011*; *Rainger et al., 2011*; *Chang et al., 1994*; *Storm and Kingsley, 1996*; *Mundy et al., 2011*; *Thomas et al., 1996*, *1997*). Joint development in the digits occurs by segmentation of existing cartilage anlagen in regions where BMP signaling is inhibited by the BMP antagonist Chordin (*Francis-West et al., 1999*) and, based on our studies, by SMOC. Once formed, the joint interzone acts as a signaling center to promote appositional growth of the opposing cartilage epipheses (*Archer et al., 2003*; *Ray et al., 2015*). Based on information from existing in vivo mutational analyses and the data presented here, we propose the following model for the role of SMOC in promoting joint formation and subsequent appositional epiphyseal growth (*Figure 8D,E*). Within the interzone, Chordin and SMOC will both bind to HSPGs (*Klemenčič et al., 2013*; *Jasuja et al., 2004*) such as syndecan-3 (*Koyama et al., 1995*), allowing diffusion of GDF5 and BMP2/4 into the epiphyses, where BMP signaling promotes chondrocyte proliferation and appositional epiphyseal growth (*Francis-West et al., 1999*; *Ray et al., 2015*) via the BMP receptor BMPR1B (*Baur et al., 2000*). This hypothesis is consistent with the skeletal and joint abnormalities observed in the absence of HS following conditional knockdown of the glycosyltransferase, Ext1, in joint interzones (*Mundy et al., 2011*). Ext1 is essential for HS synthesis and binding of Chordin to HSPGs and is necessary for its BMP inhibitory activity (*Jasuja et al., 2004*). Consequently, in the absence of HS within the interzone, the BMP antagonist activity of Chordin will be reduced. While SMOC will still contribute to the inhibition of BMP signaling, the imbalance caused by the reduced Chordin activity is consistent with the increase in BMP signaling (pSmad) observed within the interzone in the absence of HS (*Mundy et al., 2011*); as a result, in response to BMP signaling, the interzone cells undergo chondrogenesis and the joint does not form (synostosis). In the absence of SMOC, we predict that the increased availability of HSPG binding sites will restrict diffusion of GDF/BMPs out of the interzone. In addition, the imbalance in BMP inhibition within the interzone will lead to an increase in BMP signaling resulting in chondrogenesis and failure of joint formation. This hypothesis is supported by oligodactyly and joint synostosis observed in patients with Ophthalmo-Acromelic (Waardenburg Anophthalmia) Syndrome (*Okada et al., 2011*; *Rainger et al., 2011*; *Abouzeid et al., 2011*).

In conclusion, based on the data we present here together with existing literature, we propose a model to explain how vertebrate SMOC (*Thomas et al., 2009*) and *Drosophila* Pent (*Vuilleumier et al., 2010*) regulate BMP signaling. SMOC/Pent can inhibit BMP signaling locally, downstream of the BMP receptor, probably via the region containing the two Tg1-like domains. In addition, binding of SMOC/Pent to cell surface HSPGs via the C-terminal EC domain prevents BMP binding, promoting BMP diffusion, thereby expanding its morphogenetic field.

## Materials and methods

### *Drosophila* methods

UAS-*pentagone* (*UAS pent*) was constructed by PCR amplification (forward primer: ATCTCGAGCC-GAAGCACAGTAACAGTT; reverse primer: TATCTAGACGACGACATCTAATGAGTTG) from a cDNA clone corresponding to CG2264 (*Drosophila* Genomics Resource Center), and subcloned into pUAST using *XhoI* and *XbaI*. Sequence verification of the UAS-*pent* subclone followed by sequencing of the original cDNA indicated a base pair change was present in both, which resulted in a W to R amino acid change in a highly conserved region of the protein. Site directed mutatgenesis was performed with the QuikChange Lightning Site-Directed Mutagensis Kit (Stratagene) to correct *pent* cDNA to the database sequence. *Drosophila* transgenic lines were prepared by standard P-element transformation(*Spradling, 1986*) and mapped to a specific chromosome. Rescue was carried out by creating flies homozygous for the pentagone mutants: *pent$^{2-5}$/pent$^{A17}$*, and carrying the *Gal four* line *brinker -Gal 4* (*brk-Gal4*) and the UAS-*pent* construct. Adult wings were dissected, fixed in 70% ethanol,

mounted in Euperal (Asco Laboratories, Manchester, U.K.) and photographed on a Nikon E-800 microscope.

## XSMOC-1 deletion constructs for mRNA injection

XSMOC-1ΔFS (ΔQ43-A91) was created from two separate PCR fragments obtained using pCS2-XSMOC-1 (*Thomas et al., 2009*) as template. The 5' fragment was obtained using the pCS2 forward primer and the reverse primer 5'-GAGAGGATTGCACCCGGGGTCTCTGTCC-3'; the 3' fragment was obtained using the forward primer 5'-AGGTGCAAAGATCCCGGGCAGAGCAAGTGT-3' and the pCS2 reverse primer. A *Sma*I site (underlined) was incorporated to facilitate ligation of the two fragments. XSMOC-1ΔEC (ΔN310 to end) was amplified using pCS2-XSMOC-1 as template and the primer set 5'-GGCAACATGACCCCAAGA-3', 5'-CTTTAAATAGGCCTTCTCAGTCCGTATTTTTCCA-3' (incorporating a *Stu*I restriction site, underlined, and stop codon, italics). XSMOC-1ΔFS and XSMOC-1ΔEC PCR products were cloned into PCR-TOPO (Invitrogen), sequenced, and then subcloned into pCS2. XSMOC-1EC (ΔQ43 to W308), XSMOC-1Tg1 (ΔQ43-A91 and ΔN310 to end) and XSMOC-1ΔTg1 (ΔK95 to S304) in pCS2 were obtained using a primer design method to generate large deletions (*Liu and Naismith, 2008*). For XSMOC-1EC and XSMOC-1ΔTg1, the template was pCS2-XSMOC-1 and the respective primer sets 5'-CTCAGTGTTTCGGCCAAAGAGCGACTG-3', 5'-CAGTCGCTCTTTGGCCGAAACACTGAG-3' and 5'-GCAAAGATGCTGGTCAGAGCGATGCCAGATGGAAAAATACG-3', 5'-CGTATTTTTCCATCTGGCATCGCTCTGACCAGCATCTTTGC-3'. For XSMOC-1Tg1, the template was pCS2-XSMOC1ΔFS and the primer set 5'-GATGCCAGATGGAAATAGGAGAGCCGGCCAGAAG-3', CTTCTGGCCGGCTCTCCTATTTCCATCTGGCATC-3'. All pCS2 constructs were linearized with Not1 prior to mRNA transcription using the mMESSAGE mMACHINE SP6 kit (Life Technologies).

## Cloning and bacterial expression of recombinant SMOCs

Full length XSMOC-1 without the predicted signal peptide (2-24) was amplified by PCR using the forward primer 5'-TGCCATGGCCAAAGAGCGACTGGC-3' (containing a *Nco*I restriction site, underlined), the reverse primer 5'-CTCTCGAGCGCAAGGCGACTGAAGGGG T-3' (containing a *Xho*I restriction site, underlined), and full length XSMOC-1 in pCS2 (*Thomas et al., 2009*) as the template. The PCR product was cloned into PCR-TOPO (Invitrogen), sequenced, and subcloned into the pET-28b(+) expression vector (Novagen) in frame with a C-terminal hexahistidine tag. An alternative start site located within XSMOC-1 at V235 (GTG) was removed by changing the codon to GTA by site directed mutagenesis (QuikChange II kit, Agilent Technologies) using the forward primer 5'-CCAAGAGAGGGAATTGTAATTCCAGAATGTGC-3', reverse primer 5'-GCA CATTCTGGAATTACAA TTCCCTCTCTTGG-3', and XSMOC-1 in pET-28b(+) as template. Unlike XSMOC-1, the conserved Valine in hSMOC-1 is encoded by GTA in the human sequence, making the site-directed mutagenesis unnecessary. Using a primer design method to generate large deletions (*Liu and Naismith, 2008*), XSMOC-1ΔEC (ΔN310 to end) lacking the EC domain was obtained using the forward primer 5'- GATGCCAGATGG AAACACCACCACCACCACCACTG-3', the reverse primer 5'-CAGTGGTGG TGGTGGTGGTGT TTCCATCTGGCATC, and XSMOC-1 in pET-28b(+) as template. Similarly, XSMOC-1EC containing the EC domain only (ΔT2 to W308) was obtained using the same starting template, the forward primer 5'-GTGATC GGGACAGAGACAAAAATACGGACGCTGAAGACCC-3', and the reverse primer 5'-GGGTCT TCAGCGTCCGTATTTTTGTCTCTGTCCCGATCAC-3'. Several *E. coli* strains were evaluated for XSMOC-1 expression and the ShuffleT7 Express strain C3029 (New England Biolabs), which allows the formation of disulfide bonds, was found to produce the highest yield of bioactive protein. For hSMOC-1 expression the host strain was BL21DE3, as described previously (*Novinec et al., 2008*). hSMOC-1 (pET28-SMOC-1-HT(*Novinec et al., 2008*)), kindly provided by B. Lenarcic, University of Ljubljana, Slovenia, was expressed in *E. coli* strain BL21DE3. Shaker cultures (3 L) were grown at 30°C in LB broth supplemented with 30 µg/ml kanamycin. When cell densities reached $OD_{600}$ >0.5, recombinant protein expression was induced by the addition of IPTG (0.1 mM) for 5 hr. Cells were collected by centrifugation at 9000 g for 5 min.

## Refolding and purification of recombinant SMOCs

Bacterial expression and refolding was based on that described previously (*Novinec et al., 2008*) with some modifications. Bacterial pellets were washed by suspension in 40 ml of 20 mM Tris-HCl/

20% (w/v) sucrose/5 mM EDTA pH 7.5, centrifuging and resuspending in 40 ml of ice-cold deionized water. Following centrifugation the pellets were resuspended in phosphate-buffered saline (PBS)/5 mM EDTA and disrupted under high pressure (15,000 psi) using the EmulsiFlex-C3 high pressure homogenizer (Avestin, Inc, Ottawa, Canada). Lysates were centrifuged at 14,000g and inclusion bodies solubilized in 40 ml of solubilization buffer (50 mM sodium phosphate buffer, pH 8.0, 8M urea, 500 mM NaCl, 10 mM imidazole, 20 mM 2-mercaptoethanol) before applying to a pre-equilibriated 20 ml gravity-flow Ni-NTA agarose column (Qiagen). After washing with solubilization buffer bound protein was eluted with solubilization buffer/300 mM imidazole, concentrated to 5 ml (Vivaspin 20, GE Healthcare) and refolded by rapid dilution into 500 ml of refolding buffer (100 mM Tris/HCl, pH 9.0, 600 mM L-Arginine, 6 mM reduced L-Glutathione, 0.6 mM oxidized L-Glutathione, and 2 mM $CaCl_2$). Following slow stirring overnight at 4°C the refolded protein solutions were concentrated to 20 ml and dialyzed against 20 mM Tris-HCl pH 7.5, 300 mM NaCl, 2 mM $CaCl_2$. Non-soluble precipitate was removed and the dialysates concentrated to 2 ml before injecting onto a 1.6 × 60 cm HiLoad Superdex-200 PG gel filtration column (GE Healthcare) pre-equilibrated in S-200 buffer (20 mM Tris/HCl pH 7.5, 300 mM NaCl, 2 mM $CaCl_2$, 10% glycerol). Fractions containing major peaks, collected using the ÄKTA purifier 10 system and UNICORN control software (GS Healthcare), were pooled, concentrated, and analyzed by SDS-PAGE, and, when necessary, sequenced. From three separate 3L cultures the total yields of dimeric XSMOC-1 were 1.3 mg, 2.5 mg, and 5.1 mg and for dimeric XSMOC-ΔEC 3.4 mg, 4.1 mg, and 12.6 mg. The yields of monomeric XSMOC-1EC were 8.2 mg, 4.1 mg, and 3.25 mg.

## Cell culture

NIH-3T3 Fibroblasts (ATCC CRL-1658) and HEK 293 (ATCC CRL-1573) cells were cultured in DMEM medium supplemented with 10% fetal bovine serum (FBS). Prior to the addition of recombinant protein cells were serum-starved in their respective media for one hour. The stable human cervical carcinoma clonal cell line C33A-2D2, containing a multimerized BMP-responsive element (BRE) for the BMP response gene inhibitor of differentiation-1 (Id1) (*Korchynskyi and ten Dijke, 2002*) linked to luciferase was a kind gift from Martine Roussel (*Vrijens et al., 2013*). C33A-2D2 cells were maintained in Eagle's minimum essential medium (EMEM) supplemented with 10% FBS. The parental C33A cell line was purchased from ATCC (ATCC HTB-31). HEK-293 and parental C33A cells were authenticated by ATCC via STR profiling. While each cell line was confirmed to be free of mycoplasma contamination by ATCC, cells were not tested for mycoplasma in the experiments we describe. For BMP response assays, a sub-clone containing over 80% responsive cells (C33A-2D2-09) was prepared by single cell dilution. In these assays C33A-2D2-09 cells were maintained in serum-free Prime XV MSC expansion medium (Irvine Scientific) that does not contain BMPs. Wherever protein was added, XSMOC-1, XSMOC-1ΔEC, and XSMOC-1EC were used at molar equivalent amounts based on their predicted molecular weights. BMP2 (Cell Signaling #4697) was added at 50 ng/ml.

## *Xenopus* embryo manipulations

Frogs (*Xenopus laevis*), purchased from NASCO (Fort Atkinson, WI), were housed and maintained in aquaria approved by the FDA White Oak Campus Animal Care and Use Committee (ACUC). Prior to testes collection, male frogs were euthanized by anesthesia in a 2% solution of tricaine methanesulphonate, a protocol approved by the ACUC. Frog embryos were manipulated using standard methods (*Gurdon, 1967*; *Sive et al., 2000*) and euthanized by anesthesia when the required developmental stage was reached (the study was approved by the ACUC). Injection of mRNAs was performed by standard procedures as described previously (*Moos et al., 1995*). Perturbations of axial patterning were quantified by Dorso-Anterior Index (DAI, (*Kao and Elinson, 1988*)) and dark field images of embryos were photographed with low angle oblique illumination and a Zeiss Stemi-6 dissecting microscope. For animal cap assays, animal caps were removed from stage nine embryos and cultured in 0.7 x Marc's Modified Ringer's (MMR) solution (*Sive et al., 2000*), 1 mg/ml BSA/50 μg/mL gentamicin until non-injected siblings reached stage 20. Where indicated, caps were incubated in 0.7 X MMR containing XSMOC-1, XSMOC-1ΔEC, or XSMOC-1EC, at molar equivalent amounts, for different time periods.

## RT-PCR

Pools of animal cap explants from at least two different fertilizations were prepared and analyzed for each condition reported. Total RNA isolation, reverse transcription (RT), and PCR amplification were performed as described previously (*Thomas et al., 2009*). PCR products were analyzed on 1.5% agarose gels in TAE buffer, stained with SYBR Green 1 (Molecular Probes), and scanned using a Fluorimager (Molecular Dynamics).

## Immunoblotting

Cell lysates prepared by extraction in 6 M Urea, 25 mM Tris base, 2% SDS, 2% $\beta$-mercaptoethanol, and 5% glycerol were analyzed by SDS-PAGE (10 µg/lane) using Novex 10% Nu-PAGE gels (Invitrogen) and the MES buffer system. Immunoblot analyses were performed using the Novex XCell Sure-Lock Mini-Cell system (Life Technologies) and nitrocellulose membranes (Invitrogen). Transferred proteins were detected using IRDye-labeled secondary antibodies and the Odyssey infrared imaging system (Li-COR Biosciences). The primary antibodies used were, phospho-Smad 1/5/8 (Cell Signaling Technology Cat# 9511, RRID:AB_331671) and Smad1 (Cell Signaling Technology Cat #9517, RRID: AB_10699149).Peptide antibodies specific to *X*SMOC-1ΔEC (SDRDRDPQCNPHCTRPQHK) or *X*SMOC-1EC (GSFPPGKRPGSNPFSR) were produced in rabbits (Biomatik Corp.).

## Heparin-binding studies

For heparin-binding studies, 5 µg of *X*SMOC, *X*SMOCΔEC, *X*SMOC-EC or human BMP2 (R and D Systems) in 50 µL of 1X PBS/0.5M NaCl was added to 20 µL of pre-equilibrated heparin Sepharose high performance beads (GE Life Sciences) and mixed with rotation for 15 min at room temperature. The beads were centrifuged (350 g for 2 min) and the supernatant removed. The protein-heparin bead mixture was then washed twice with 500 µL of 1X PBS or 1X PBS/NaCl (0.4 to 0.7M) before elution with 20 µL of 1 X Lithium Dodecyl Sulphate (LDS) sample buffer (Invitrogen)/0.1 M DTT for 5 min at 95°C. The supernatants were analyzed on a 10% NuPAGE gel and visualized by Coomassie staining.

## BMP in vitro diffusion assay

Affi-Gel Blue beads (BioRad) approximately 100 µm in diameter were soaked in 100 µg/ml BMP4 (R and D Systems # 314 BP-010/CF) for 3 hr in a humidified chamber. Drops (0.5 µl) of 0.7% low melting point agarose (Life Technologies) with or without heparan sulfate (10 µg/ml; Sigma Aldrich #H7640) and either *X*SMOC-1EC (100 µg/ml) or S-200 buffer were placed on Millicell EZ chamber slides (Millipore). The slides were placed in a humidification chamber and incubated at room temperature for 5 min. Individual BMP4-soaked beads were placed into the partially gelled matrices and incubation was continued until gelling was complete. C33A-2D2-09 cells ($2 \times 10^4$) in Prime XV MSC expansion medium (400 µl) were seeded into each well. The slides were incubated at 37°C in 5% $CO_2$ for 48 hr prior to detection of luciferase. Briefly, following fixation (15 min) in 4% paraformaldehyde/PBS, cells were washed and permeabilized (0.5% Triton-X-100/PBS) for 10 min. Slides were washed (TBS/0.05% Tween) and incubated in blocking solution (Duolink, Olink Biosciences) for 1 hr in a humidified chamber prior to incubation with goat anti-luciferase (20 µg/ml) primary antibody (Promega #G745A, RRID:AB_2335880) for 1 hr. After washing (Duolink Wash Buffer A), slides were incubated in donkey anti-goat Alexa-488 (Thermo Fisher Scientific Cat# A-11055, RRID:AB_2534102) secondary antibody (5 µg/ml) for 45 min. Slides were then washed three times each in Duolink Wash Buffers A and B, mounted in Duolink mounting medium with DAPI, imaged by confocal microscopy (Zeiss LSM710), and analyzed with ImageJ (NIH) software.

## BMP in vivo diffusion assay

Each blastomere of *Xenopus* embryos at the two cell stage was injected with mRNAs for BMP2 (15 pg or 150 pg), or SMOC (5, 15, or 150 pg) and/or mCherry (50 pg). pCS2 +8 NmCherry (*Gökirmak et al., 2012*) was a gift from Amro Hamdoun (Addgene plasmid # 34936). Animal caps were removed at stage nine and grafted into the animal poles of control or BMP2-injected embryos from which the animal poles had been removed. Following a two hour incubation the animal halves of the embryos were removed and fixed for 25 min in PBS/4% paraformaldehyde. The explants were permeabilized in 0.5% TritonX-100 for 10 min, washed in PBS, then transferred to blocking solution

(PBS/5% goat serum) for 1 hr before incubating overnight at 4°C in a 1/1500 dilution of anti-pSMAD 1/5/8 antibody (Cell Signaling Technology Cat# 9511, RRID:AB_331671). The tissue was washed in PBS and prepared for immunofluorescence by incubating in goat anti-rabbit Alexa 488 (Thermo Fisher Cat#A11008, RRID:AB_143165) secondary antibody (5 µg/ml) for 2 hr. The tissue was then washed three times in PBS, transferred to a glass slide, mounted in Duolink mounting medium with DAPI, and imaged by confocal microscopy (Zeiss LSM710).

## Acknowledgements

We thank Giorgos Pyrowolakis for *Drosophila* stocks, Martine Roussel for the stable human cervical carcinoma clonal cell line C33A-2D2, and Brigita Lenaričič for the hSMOC-1 expression vector, pET28-SMOC-1-HT. The *Drosophila* Genomics Resource Center, supported by NIH grant 2P40OD010949-10A1, provided the *pentagone* cDNA clone. The constitutively active BMP receptor construct (caBMPR1B) was the kind gift of Lee Niswander. We thank Howard Mostowski for his expert help with flow cytometry and are indebted to the staff of the FDA/CBER Division of Veterinary Services for their superb care of our *Xenopus* colony. We thank Ian Bellayr and Bharat Joshi for internal review of the manuscript.

## Additional information

### Funding

| Funder | Author |
|---|---|
| U.S. Department of Health and Human Services | J Terrig Thomas<br>Kristin R Andrykovich<br>Brian G Stultz<br>Deborah A Hursh<br>Malcolm Moos |
| FDA Commissioner's Fellowship Program | D Eric Dollins<br>Tehyen Chu |

The funders had no role in study design, data collection and interpretation, or the decision to submit the work for publication.

### Author contributions

JTT, Conceptualization, Data curation, Formal analysis, Supervision, Investigation, Methodology, Writing—original draft, Writing—review and editing; DED, BGS, Data curation, Formal analysis, Investigation, Methodology, Writing—review and editing; KRA, Data curation, Writing—review and editing, investigation; TC, Data curation, Formal analysis, Writing—review and editing; DAH, Conceptualization, Data curation, Formal analysis, Investigation, Methodology, Writing—review and editing; MM, Conceptualization, Formal analysis, Supervision, Methodology, Writing—original draft, Writing—review and editing

### Author ORCIDs

J Terrig Thomas, http://orcid.org/0000-0001-9252-1011

### Ethics

Animal experimentation: Frogs (Xenopus laevis) were housed and maintained in aquaria approved by the FDA White Oak Campus Animal Care and Use Committee (ACUC). Prior to testes collection, male frogs were euthanized by anesthesia in a 2% solution of tricaine methane-sulphonate, a protocol approved by the ACUC. Frog embryos were manipulated using standard methods and euthanized by anesthesia when the required developmental stage was reached (the study was approved by the ACUC). The ACUC approved protocol associated with this work is 1989-129

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
