## [Decision Letter]

Thank you for submitting your article "SMOC can act as both an antagonist and an expander of BMP signaling" for consideration by *eLife*. Your article has been reviewed by two peer reviewers, and the evaluation has been overseen by Janet Rossant as the Senior Editor and Hugo J Bellen as the Reviewing Editor. The reviewers have opted to remain anonymous.

The reviewers have discussed the reviews with one another and the Reviewing Editor has drafted this decision to help you prepare a revised submission.

Summary:

The manuscript by Thomas et al. focuses on *Xenopus* Smoc-1- a BMP modulator with negative effects onto BMP signaling. Previously, the authors uncovered a role for Smoc in MAPK- dependent phosphorylation of the Smad linker, thus an indirect entry into the BMP signaling. However, data from vertebrates and *Drosophila* argue for a role for Smoc in modulating BMPs availability outside the cells. For example, the *Drosophila* Pent, appears to expand the range of BMP signaling and to promote signaling at a distance. In this manuscript, the authors test whether Smoc has dual activities onto the BMP signaling and address how different functional domains of Smoc inhibit BMP signaling at short-range while promoting long-range signaling. The authors take a biochemical approach and generate recombinant Smoc variants and refold biologically active proteins. Using these variants in multiple and creative experimental settings, they conclude that the EC domain of Smoc mediates its binding to glypicans and competes with BMPs for glypicans occupancy. From the data presented, it is less clear whether Smoc domain(s) could mediate glypican-independent inhibition of BMP signaling.

In order to understand the dual role of SMOC as an antagonist of BMP signalling and an expander of BMP spreading, the authors establish a protocol to express and refold recombinant wild-type (wt) and truncated versions of SMOC. Using recombinant wt and truncated SMOC, Thomas et.al. demonstrate that the C-terminal heparin sulphate(HS)-binding domain is dispensable for cell fate conversion and SMOC-mediated repression of BMP signalling. In contrast, the SMOC HS-binding domain is sufficient to promote BMP signalling and to compete with BMP2 for heparin sepharose binding. Furthermore, the authors show that competition for HSPG-binding between BMP2 and the HS-binding domain of SMOC can result in increasing range of BMP2 signalling in a cell culture assay. Based on the finding that SMOC potentially competes with BMPs for HSPG binding, the authors suggest a model for SMOC/Pent function.

The major novelty of this manuscript is the finding that SMOC can compete with BMP ligands for binding to heparin sulphates. Hence, these results suggest a novel mechanism by which SMOC/Pent would regulate the spreading of BMP ligands by competitive binding to extracellular HSPG.

The reviewers have some issues with the paper that need to be addressed, but which they believe can be addressed within the required 2-month time window.

The conclusions of this study are based on in vitro essays only, hence it does not provide evidence that SMOC indeed competes with BMPs under physiological conditions for binding with HSPG. Therefore providing in vivo data that supports the suggested mechanism would greatly enhance the conclusion of this manuscript. If providing such data is not possible, the authors should at least provide in vivo evidence that SMOC can indeed promote BMP signalling at a distance. This might be done by implanting SMOC-soaked beads close to a BMP source and monitor the effect on BMP signalling range.

While this is undoubtedly an important developmental question, this manuscript lacks several important controls and tends to over-interpret the structure-function analyses. Several major concerns need to be addressed before consideration for publication:

1) First, the authors argue that Smoc and delta-EC have equivalent effects, but from the data (without quantification) it seems that delta-EC has a reduced effect compared with full-length Smoc, for example in Figure 3, Figure 4. This result needs to be clarified before the authors could uncouple the glypican-binding function of Smoc from other activities.

2) The data presented in Figure 4 are over-interpreted and do not demonstrate a role for the Tg domains in the inhibition of BMP signaling. The arguments regarding the role of EC in promoting BMP signaling are convincing, but the Tg experiment in 4D is incomplete and should present a full set of relevant Smoc and co-Pent truncations, including delta-FS and delta-EC together, as well as delta-Tg. The 4D panel is based on capped mRNA microinjection (and not purified and refolded proteins), thus a comprehensive set of truncations should be feasible and must be analyzed, quantified, and included here.

3) Figure 6 shows that EC, and not Smoc, can promote BMP signaling at a distance. Experiments with full-length Smoc are not shown or described.

This experiment is very creative and circumvents a tough technical problem in the field, the stickiness of BMP-type ligands. (Since BMP2 is very sticky, Smoc is unlikely to displace it from Heparin-Sepharose, so the converse experiment for Figure 5 would not work.) Setting up a cell-based system to examine whether Smoc displaces BMP from HSPSs is a clever alternative. But it is unfortunate that the authors did not present the entire series, including Smoc, and delta-EC.

Importantly, better and more uniform fields of cells must be chosen to illustrate this result, since more cells (as apparent in 6B) will deplete the BMP2 pool faster than the fewer cells shown in 6C, muddying the data.

4) How do these new findings correlate with the previous report from this group that Smoc modulates MAPK-dependent Smad phosphorylation? The authors need to include a section in the Discussion in which their current results should be reconciled and tied together with their previous findings.

The recent *Drosophila* Pent paper (Norman et al., *eLife* 2016), which shows that Pent also modulates the Wg/Wnt signaling via glypicans, together with previous results reported by Eddy De Robertis' group (Fuentealba et al., Cell 2007) may offer some explanations. For example, the interaction between Pent and Glypicans may play a key role in integrating patterning signals by controlling the interplay between MAPK, Wnt/GSK3 and BMP signaling, which seem to converge onto the phosphorylation state and duration of Smad-dependent signaling.

---

## [Author Response]

*[…] The reviewers have some issues with the paper that need to be addressed, but which they believe can be addressed within the required 2-month time window.*

*The conclusions of this study are based on in vitro essays only, hence it does not provide evidence that SMOC indeed competes with BMPs under physiological conditions for binding with HSPG. Therefore providing in vivo data that supports the suggested mechanism would greatly enhance the conclusion of this manuscript. If providing such data is not possible, the authors should at least provide in vivo evidence that SMOC can indeed promote BMP signalling at a distance. This might be done by implanting SMOC-soaked beads close to a BMP source and monitor the effect on BMP signalling range.*

We agree with the reviewer’s comment that data in vivo would greatly enhance our conclusions. We have designed and conducted experiments using *Xenopus* embryos to address this issue, the results of which are presented in a new Figure 7.

*Xenopus* embryos were injected at the two cell stage with mRNAs for BMP2, or SMOC and/or mCherry. Animal caps (ACs) from SMOC/mCherry-injected embryos (donor) were removed at stage 9 and conjugated to the animal poles of BMP2-injected embryos from which ACs had been removed (host). After two hours the entire animal hemispheres of the host/donor conjugates were fixed and immunostained for pSmad 1/5/8 to detect BMP signaling. mCherry was used to distinguish the SMOC-injected donor tissue from the BMP2-expressing host tissue. The use of ACs in this assay was supported by the observation that endogenous pSmad signaling is not present in stage 9 to 9.5 ACs (Faure et al., 2000, Development 127, 2917-2931). As background pSmad staining was absent in the control conjugates, cells positive for pSmad in the donor AC tissue would be indicative of diffusion of BMP from the surrounding host tissue.

A number of mRNA titration experiments were conducted to determine the optimal amounts of mRNA to inject and the time to incubate the conjugates prior to immunostaining for pSmad. It was found that injection of *Xenopus* embryos at the two cell stage with 30pg BMP2 mRNA was sufficient to obtain pSmad staining in the host tissue, but insufficient diffuse into the donor Wt caps after two hours. When higher amounts of BMP2 mRNA were used (300pg) pSmad was detected throughout the donor AC tissue, demonstrating that two hours was sufficient time for BMP to diffuse into the donor AC and induce pSmad. The ability of SMOC to extend the range of BMP signaling was tested by conjugating donor ACs from *Xenopus* embryos injected with different amounts SMOC mRNA to hosts injected with 30pg of BMP2. We found that much lower amounts of SMOC mRNA were required (10 to 30pg) than that used previously to inhibit BMP signaling (300pg). When high amounts of SMOC mRNA (300pg) were used, there was no pSmad staining in the donor AC or within the first few cell diameters of the adjacent host tissue, indicating BMP inhibition. The results of these experiments have been incorporated into the revised manuscript and a new figure (Figure 7) is provided.

We hope that this assay and the data obtained will be sufficient to address the reviewers’ request to show the ability of SMOC to promote BMP signaling at a distance from its source in vivo.

A question we have for the reviewers is whether including hatch marks to demarcate the host/donor tissue boundaries would be beneficial. We did not include these in the current figure as we did not want to obscure the data in any way.

*While this is undoubtedly an important developmental question, this manuscript lacks several important controls and tends to over-interpret the structure-function analyses. Several major concerns need to be addressed before consideration for publication:*

*1) First, the authors argue that Smoc and delta-EC have equivalent effects, but from the data (without quantification) it seems that delta-EC has a reduced effect compared with full-length Smoc, for example in Figure 3, Figure 4. This result needs to be clarified before the authors could uncouple the glypican-binding function of Smoc from other activities.*

With regard to Figure 3, while we do not state or intend to imply that SMOC and SMOC-∆EC are fully equivalent. However, we do show and state that both are able to induce the expression of neural markers in *Xenopus* animal caps (subsection “BMP Inhibition and Neural induction by SMOC does not require the EC domain”, first paragraph). Similarly, Figure 4, amended to show the effect of each deletion construct, clearly shows that SMOC and SMOC∆EC can inhibit BMP signaling.

*2) The data presented in Figure 4 are over-interpreted and do not demonstrate a role for the Tg domains in the inhibition of BMP signaling. The arguments regarding the role of EC in promoting BMP signaling are convincing, but the Tg experiment in 4D is incomplete and should present a full set of relevant Smoc and co-Pent truncations, including delta-FS and delta-EC together, as well as delta-Tg. The 4D panel is based on capped mRNA microinjection (and not purified and refolded proteins), thus a comprehensive set of truncations should be feasible and must be analyzed, quantified, and included here.*

Two additional SMOC truncation constructs were prepared; SMOC-∆FS∆EC (referred to as SMOC-Tg1), containing the signal peptide and the two Tg1-like domains separated by the non-homologous domain, and SMOC-∆Tg1, containing the FS- like domain and EC domains (subsection “*X*SMOC-1 deletion constructs for mRNA injection”). We conducted mRNA overexpression studies in *Xenopus* embryos with SMOC, SMOC∆FS, SMOC∆EC, SMOC-Tg1 only (SMOC∆FS∆EC), and SMOC-∆Tg1 in the presence of the constitutively active BMP receptor (caBMPR1B) to test their ability to inhibit BMP signaling. Overexpression of SMOC, SMOC-∆FS, and SMOC-∆EC inhibited BMP signaling in the presence of the caBMPR; SMOC-Tg1, SMOC-∆Tg1 and SMOC-EC did not (subsection “BMP Inhibition and Neural induction by SMOC does not require the EC domain”, last paragraph). We have replaced the original Figure 4 panel D with the RT-PCR results obtained from these experiments. We speculate, by process of elimination, that while we believe the BMP inhibition activity of SMOC resides in the Tg1 domains, the protein translated from the SMOC-Tg1 only deletion construct may be inactive due to misfolding. We have deleted any categorical statements we had made previously regarding the involvement of the Tg1 domains and replaced them with more speculative language. In addition, we haveincluded data from human genetic studies that support the involvement of the SMOC Tg1 domains in SMOC function (see the aforementioned paragraph).

We did not prepare truncation constructs for co-Pent as, although the domain structuresof SMOC and Pent are similar, there are also significant differences that would make truncation comparisons difficult to interpret. Consequently, apart from showing that SMOC and co-Pent can both inhibit BMP signaling, we feel this request is beyond the scope of this manuscript.

3) Figure 6 shows *that EC, and not Smoc, can promote BMP signaling at a distance. Experiments with full-length Smoc are not shown or described.*

*This experiment is very creative and circumvents a tough technical problem in the field, the stickiness of BMP-type ligands. (Since BMP2 is very sticky, Smoc is unlikely to displace it from Heparin-Sepharose, so the converse experiment for Figure 5 would not work.) Setting up a cell-based system to examine whether Smoc displaces BMP from HSPSs is a clever alternative. But it is unfortunate that the authors did not present the entire series, including Smoc, and delta-EC.*

*Importantly, better and more uniform fields of cells must be chosen to illustrate this result, since more cells (as apparent in 6B) will deplete the BMP2 pool faster than the fewer cells shown in 6C, muddying the data.*

Having shown that the EC domain is required for binding to heparan sulfate and has the same affinity as full length SMOC (Figure 5), this in vitro assay was designed to show that HS-binding by the EC domain of SMOC promotes BMP diffusion through a HS-containing matrix. We did not assess the effect of full length SMOC or SMOC-∆EC in this assay as both could also potentially inhibit BMP signaling and mask the effect on BMP diffusion.

With regard to the comment on the fields of cells chosen to illustrate the results, we have quantified the percentage of luciferase-positive cells in each field obtained over four separate experiments and included this data in the figure legend. We consider the panels shown to be representative of the experimental results obtained for this assay and have stated this in the figure legend.

*4) How do these new findings correlate with the previous report from this group that Smoc modulates MAPK-dependent Smad phosphorylation? The authors need to include a section in the Discussion in which their current results should be reconciled and tied together with their previous findings.*

A discussion of how our previous work on BMP inhibition by SMOC involving linker Smad phosphorylation and our current results has been included (subsection “Model to explain the dual function of Pent and SMOC as BMP antagonists and expanders of BMP signaling”, last paragraph)

*The recent Drosophila Pent paper (Norman et al., eLife 2016), which shows that Pent also modulates the Wg/Wnt signaling via glypicans, together with previous results reported by Eddy De Robertis' group (Fuentealba et al., Cell 2007) may offer some explanations. For example, the interaction between Pent and Glypicans may play a key role in integrating patterning signals by controlling the interplay between MAPK, Wnt/GSK3 and BMP signaling, which seem to converge onto the phosphorylation state and duration of Smad-dependent signaling.*

A discussion of the findings presented in the Norman paper and how these relate to our data has been included (subsection “Model to explain the dual function of Pent and SMOC as BMP antagonists and expanders of BMP signaling”, second paragraph).